# CAN SMALL TRAINING RUNS RELIABLY GUIDE DATA CURATION? RETHINKING PROXY-MODEL PRACTICE

**Jiachen T. Wang**
Princeton University

**Tong Wu**
Princeton University

**Kaifeng Lyu**[*]
Tsinghua University

**James Zou**
Stanford University

**Dawn Song**
UC Berkeley

**Ruoxi Jia**[†]
Virginia Tech

**Prateek Mittal**[†]
Princeton University

## ABSTRACT

Data teams at frontier AI companies routinely train small proxy models to make critical decisions about pretraining data recipes for full-scale training runs. However, the community has a limited understanding of whether and when conclusions drawn from small-scale experiments reliably transfer to full-scale model training. In this work, we uncover a subtle yet critical issue in the standard experimental protocol for data recipe assessment: the use of identical small-scale model training configurations across all data recipes in the name of "fair" comparison. We show that the experiment conclusions about data quality can flip with even minor adjustments to training hyperparameters, as the optimal training configuration is inherently data-dependent. Moreover, this fixed-configuration protocol diverges from full-scale model development pipelines, where hyperparameter optimization is a standard step. Consequently, we posit that the objective of data recipe assessment should be to identify the recipe that yields the best performance under data-specific tuning. To mitigate the high cost of hyperparameter tuning, we introduce a simple patch to the evaluation protocol: using *reduced* learning rates for proxy model training. We show that this approach yields relative performance that strongly correlates with that of fully tuned large-scale LLM pretraining runs. Theoretically, we prove that for random-feature models, this approach preserves the ordering of datasets according to their optimal achievable loss. Empirically, we validate this approach across 23 data recipes covering four critical dimensions of data curation, demonstrating dramatic improvements in the reliability of small-scale experiments.

## 1 INTRODUCTION

High-quality data has emerged as the primary driver of progress in modern AI development Comanici et al. (2025). Constructing the data recipe for training frontier AI models requires a series of high-stakes decisions—such as how to filter low-quality text, how aggressively to deduplicate, and how to balance diverse data sources. Unfortunately, there is little theoretical guidance or human intuition to direct these choices, leaving practitioners to rely on actual model training for data quality assessment.

**Proxy-model-based techniques.** The most direct approach to selecting a data recipe is to train full-scale models for each candidate recipe and compare their performance. However, this is prohibitively expensive for large-scale model training, as it would require numerous complete training runs. Researchers and practitioners have widely adopted smaller "proxy models" as surrogates to efficiently estimate each dataset's utility for full-scale training, significantly reducing the computational burden (Coleman et al., 2020). Given the computational efficiency and ease of implementation, small-scale proxy model experiments have guided data decisions for many high-profile models and open-source datasets (Schuhmann et al., 2021; Mazumder et al., 2023; Li et al., 2024; Dubey et al., 2024).

**Rethinking data recipe ablation in practical workflows.** Given the critical importance of data quality, modern AI development teams typically employ a division of labor: *specialized data teams*

---

[*]Work done while at UC Berkeley.
[†]Equal contribution as senior authors.

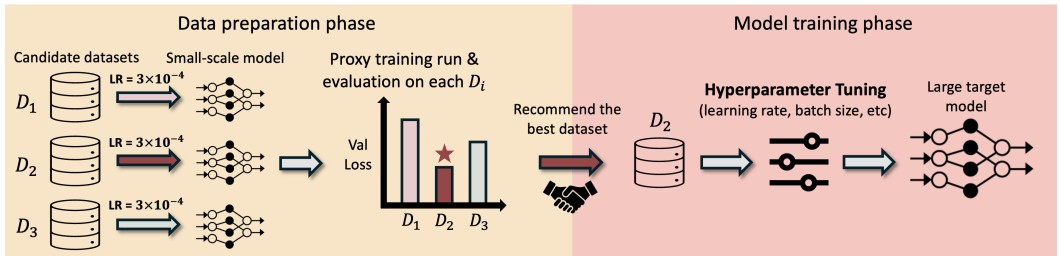

Figure 1: Overview of the industrial practice of data recipe ablation using proxy models. **Left**: Data teams assess each candidate dataset using small-scale proxy models trained with *identical* hyperparameters, and recommend the data recipe with the best performance. **Right**: Model training teams perform extensive hyperparameter tuning to optimize performance on the large target model.

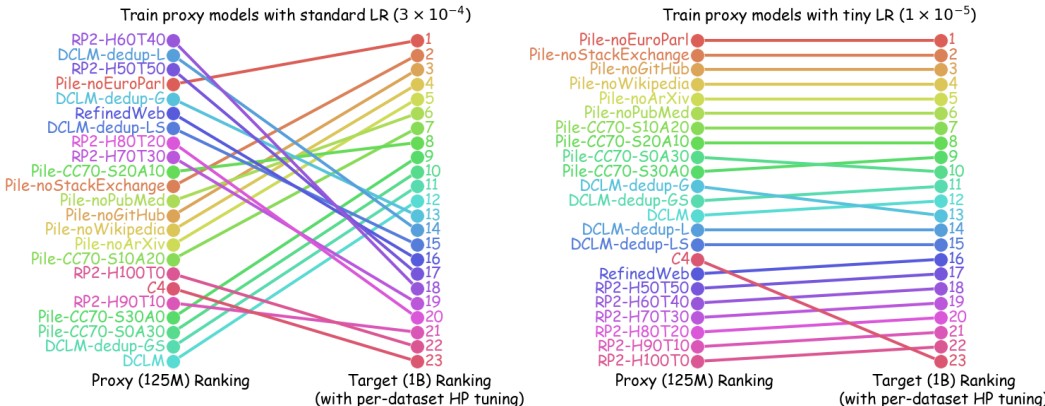

Figure 2: Validation loss rankings of 23 data recipes evaluated on proxy models (GPT2-125M) and target models (Pythia-1B), where target models undergo extensive dataset-specific hyperparameter tuning. Rankings are determined by the pretrained models' loss on Pile's validation split (Gao et al., 2020). **Left:** When proxy models are trained with a standard learning rate ($3 \times 10^{-4}$), data recipe rankings exhibit severe disagreement between proxy and target scales, with many dramatic reorderings that would lead to suboptimal data recipe ablation. **Right:** When proxy models are trained with a tiny learning rate ($1 \times 10^{-6}$), dataset rankings remain highly consistent across scales.

curate and optimize training data recipe, then recommend a high-quality dataset to *model training teams* who subsequently optimize training configurations (e.g., hyperparameters) specifically for the dataset (Hoffmann et al., 2022; Anil et al., 2023; Almazrouei et al., 2023) (Figure 1). However, existing data-centric research evaluates and compares data recipes in ways fundamentally disconnected from this practical workflow. Most experiments in the data-centric literature and benchmarks evaluate all candidate data recipes under *fixed* training hyperparameters for "fairness", while practical model training uses *optimally-tuned hyperparameter configurations specifically adapted to each dataset*. We therefore propose a refined objective for data-centric AI: find the data recipe that maximizes performance under *optimally-tuned hyperparameters*, reflecting how data is actually used in practice.

**Minor variations in proxy training configuration can alter data recipe rankings (Section 4).** The refined objective imposes a critical requirement for proxy-model-based techniques to succeed: the best data recipe identified through small-scale training runs must remain superior after (i) scaling the model to its target size and (ii) the training team optimizes training hyperparameters. However, this goal presents significant challenges given current proxy-model-based methods, where small proxy models evaluate and rank candidate datasets under a fixed, heuristically chosen hyperparameter configuration. Due to the strong interdependence between training hyperparameters and data distribution, each data recipe inherently requires its own optimal training configuration (Hutter et al., 2019; Sivaprasad et al., 2020). Our experiments (Figure 3) demonstrate that in language model pretraining, even minor variations in training hyperparameters, particularly learning rate, can dramatically alter data recipe

rankings and lead to suboptimal recommendations. If a dataset ordering collapses under such minor hyperparameter tweaks at the same proxy scale, broader sweeps at larger scales will almost certainly reorder the ranking further.

**Training proxy models with tiny learning rates: a theoretically-grounded patch (Section 5).** In the long run, the tight interplay between data and training hyperparameters suggests that these two components should be optimized *jointly* rather than tuned in separate, sequential steps. In the meantime, however, practitioners still need a *drop-in patch* that allows data teams to assess and optimize their data curation pipelines through small-scale experiments alone. We propose a simple yet effective remedy: train the proxy model with a *tiny* learning rate. Two key empirical observations inspire this approach: (i) within the same model architecture, datasets' performance under a tiny learning rate strongly correlates with their optimal achievable performance after dataset-specific hyperparameter tuning; (ii) data recipe rankings remain stable under tiny learning rates as models scale from small to large sizes. We provide formal proof for these empirical findings for random feature models. Specifically, we prove that, as network width grows, training with a sufficiently small learning rate preserves the ordering of datasets, and the rankings converge to the ordering of their *best achievable performance* in the infinite-width limit. We further characterize this tiny learning rate regime through both theory and practical heuristics.

**Experiments.** We empirically evaluate the effectiveness of the tiny-learning-rate strategy through comprehensive experiments spanning multiple architectures, scales, and data curation scenarios. We test 23 data recipes covering domain composition, quality filtering, deduplication strategies, and major pretraining corpora across three language model families (GPT2, Pythia, OPT), ranging from 70M to 1B parameters. Our results demonstrate that proxy models trained with small learning rates achieve significantly improved transferability to *hyperparameter-tuned* larger target models. Figure 2 illustrates this improvement: the Spearman rank correlation for data recipe rankings between GPT2-125M and Pythia-1B improved to $> 0.95$ across $\binom{23}{2} = 253$ data recipe pairs when using a tiny learning rate ($10^{-5}$) to train GPT2 instead of a commonly used learning rate ($3 \times 10^{-4}$).

## 2 Background: Guiding Data Curation with Small Proxy Models

In this section, we formalize the problem of data recipe ablation for large-scale model training.

**Setup & Notations.** Consider a target model architecture $\theta_{\text{tgt}}$ and a pool of candidate datasets $\mathcal{D} = \{D_1, D_2, \ldots, D_n\}$, where each dataset results from a different *data recipe* (e.g., different curation algorithms, filter thresholds, or domain mixing ratios). Data recipe ablation aims to identify the optimal dataset $D_{i^*} \in \mathcal{D}$ that maximizes model performance on a validation set $D_{\text{val}}$. Given a loss function $\ell$, let $\ell_{\text{val}}(\theta) := \ell(\theta; D_{\text{val}})$ denote the validation loss of model $\theta$. Since model performance depends critically on both training data and hyperparameters (e.g., learning rate, batch size), we write the trained model as $\theta(D; \lambda)$ for dataset $D$ and hyperparameter configuration $\lambda$.[1]

**Current practice of data recipe ablation with small proxy models.** Directly evaluating candidate datasets by training a large target model on each is typically computationally prohibitive. A common practice to mitigate this issue involves using smaller "proxy models" ($|\mathcal{M}_{\text{proxy}}| \ll |\mathcal{M}_{\text{target}}|$) to predict data quality and determine which data curation recipe to use in large-scale training runs. Small models enable repeated training at substantially reduced computational costs, making them popular for ablation studies of various data curation pipelines (Dubey et al., 2024; Li et al., 2024). Current practices typically train proxy models on each $D_i$ (or its subset) using a *fixed* hyperparameter configuration $\lambda_0$, then rank the datasets based on small models' performance $\ell_{\text{val}}(\mathcal{M}_{\text{proxy}}(D_i; \lambda_0))$.

## 3 Rethinking "High-quality Dataset": A Practical Development Perspective

Despite the broad adoption of small proxy models for data recipe ablation, the research community has a limited understanding of the conditions under which conclusions from small-scale experiments

---

[1]Additionally, the training outcome is influenced by stochastic elements such as random initialization and data ordering. We omit the randomness here for clean presentation and treat $\theta(\cdot)$ as deterministic. We provide additional discussion on the impact of stochasticity on proxy-based selection in Appendix D.2.1.

can be reliably transferred to large-scale production training. Before delving into the question of whether datasets that appear superior for small model training remain optimal for larger models, we must first establish a principled objective for data quality assessment. That is, *what constitutes a "high-quality dataset" in practical model development?* In this section, we discuss a subtle yet critical disconnect between practical workflows and the standard evaluation protocol in data-centric research.

**Data recipes need to be assessed under individually optimized training configurations.** In the existing literature (e.g., Magnusson et al. (2025)), the effectiveness of data-centric algorithms is typically assessed by training a large target model on the curated dataset with a *fixed* set of hyperparameters. However, in the actual AI development pipelines, hyperparameter tuning will be performed, and the hyperparameters are *tailored to the curated dataset*. For instance, GPT-3 (Brown et al., 2020) determines the batch size based on the gradient noise scale (GNS) (McCandlish et al., 2018), a data-specific statistic. Similarly, learning rates (Yang et al., 2022) and optimization algorithms (Chen et al., 2022) are usually adjusted in a data-dependent way. Therefore, we argue that a more reasonable goal for the data recipe ablation should optimize the selected dataset's performance under *hyperparameters tuned specifically for that dataset*. This refined objective acknowledges the strong interaction between data and training hyperparameters in model training (Hutter et al., 2019), emphasizing that data curation strategies should optimize for the training data's *optimally achievable performance* rather than performance under a predefined, potentially suboptimal hyperparameter configuration. Formally, we formalize the data recipe ablation problem as $D_{i*} := \arg\min_{i \in [n]} \min_{\lambda \in \Lambda} \ell_{\text{val}}(\theta(D_i; \lambda))$, where $\Lambda$ is a predefined feasible space of hyperparameters constrained by factors such as the available compute budget.

## 4 PROXY-MODEL FRAGILITY UNDER HYPERPARAMETER VARIATION

Given our refined objective of identifying the dataset that performs best under its own optimized hyperparameters, we now examine whether current proxy-model practices align with this goal. Standard practice evaluates each candidate dataset by training a proxy model with a single, heuristically chosen hyperparameter configuration. Our investigation reveals a concerning consequence: even minor adjustments to the learning rate can flip the conclusions drawn from small proxy training runs. Consequently, the current practices can fail to identify datasets with the highest potential even for the same proxy model, and are even more likely to select suboptimal datasets when scaled to larger models with proper hyperparameter tuning. Through sweeps of critical training hyperparameters—including batch size, weight decay, and token-per-parameter ratios—we find that dataset rankings are most sensitive to learning rate. Consequently, we center our main analysis on learning rate and detail the remaining hyperparameter experiments in Appendix D.2.2.

**Experiments: Learning rate sensitivity can undermine proxy models' reliability.** We investigate how minor variations in learning rate values can flip the conclusions drawn from small-scale experiments. Figure 3 demonstrates this fragility by comparing DCLM (Li et al., 2024) with one of its variants that underwent more stringent deduplication (detailed configurations in Appendix D.1.4).

We train GPT2-Small (125M) on each dataset using two similar learning rates that reflect typical choices made by practitioners (Karpathy, 2022). The results reveal that dataset rankings can be inconsistent with respect to the chosen learning rate. At a lower learning rate, DCLM is superior on both the validation loss and the downstream benchmarks. However, a slight increase in learning rate reverses this performance ranking. This reversal likely stems from DCLM's relatively loose deduplication criteria, which tend to favor smaller learning rates, whereas its more stringently deduplicated variant performs better with larger learning rates. The sensi-

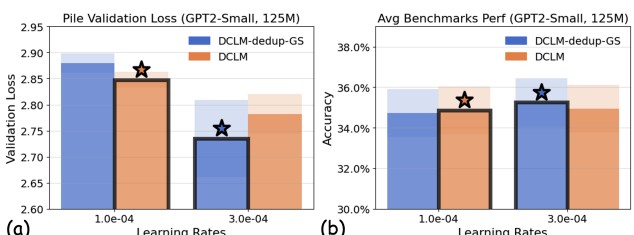

Figure 3: Performance comparison of DCLM variants by training GPT2-Small with two similar learning rates in a near-optimal regime. DCLM-dedup-GS is a variant of the DCLM dataset constructed with more stringent deduplication thresholds (see Table 1 for details). **(a)** Validation loss on the Pile dataset. **(b)** Average accuracy across 5 downstream benchmarks (see Section 6.1). Star markers indicate the superior recipe in each setting. The shaded regions represent the standard errors calculated across 3 random seeds.

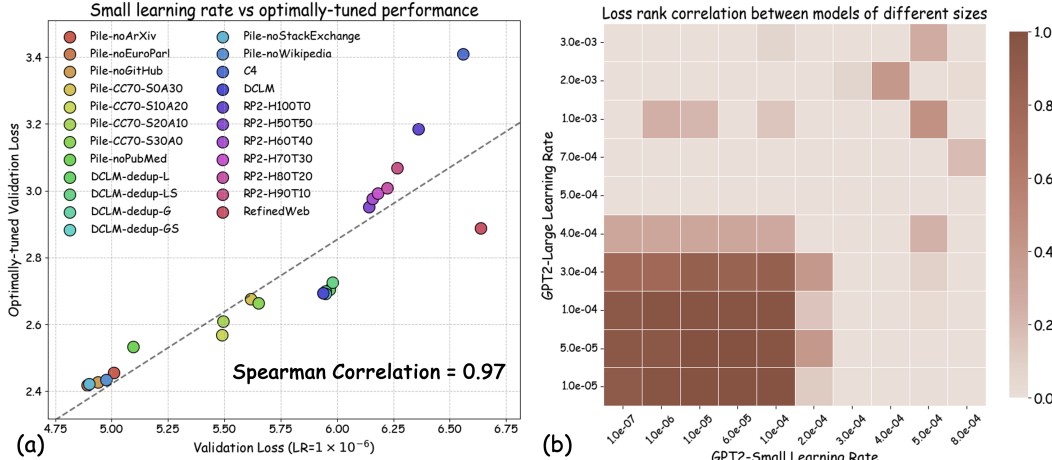

Figure 4: **(a)** Correlation between losses of GPT2-Small trained with a small learning rate $1 \times 10^{-6}$ versus optimally tuned hyperparameters, evaluated on Pile's validation split. Each point represents one dataset, demonstrating that tiny learning rate performance is strongly correlated with the optimal performance for the same model architecture. **(b)** Heatmap showing Spearman rank correlation between dataset rankings from proxy (GPT2-Small) and target (GPT2-Large) models trained with varying learning rate combinations. High correlation (darker area) in the lower-left indicates that when both models use tiny learning rates, dataset rankings are preserved across model scales.

tivity to learning rate selection high-
lights a critical limitation of small-scale proxy experiments: conclusions drawn from small-scale training runs with *fixed* configurations can overfit to those specific settings, potentially leading to suboptimal data curation decisions when scaling to large-scale model training.

**High-level intuition: The curse of higher-order effects.** To gain intuition about why different learning rates lead to inconsistent data recipe rankings in proxy models, we consider a highly simplified one-step gradient descent setting, where the change in validation loss $\ell_{\mathrm{val}}$ after one gradient update can be approximated through Taylor expansion:

$$\Delta\ell_{\mathrm{val}}(\theta) = \ell_{\mathrm{val}}(\theta - \eta\nabla\ell(\theta)) - \ell_{\mathrm{val}}(\theta) \approx -\eta\nabla\ell_{\mathrm{val}}(\theta) \cdot \nabla\ell(\theta) + \frac{\eta^2}{2}\nabla\ell(\theta)^T H_{\ell_{\mathrm{val}}}(\theta)\nabla\ell(\theta)$$

where $H_{\ell_{\mathrm{val}}}$ is the Hessian of the validation loss. Consider two datasets $D_i$ and $D_j$ with corresponding training losses $\ell_i$ and $\ell_j$. When the learning rate is low, their ranking primarily depends on the first-order gradient alignment terms: $\nabla\ell_{\mathrm{val}}(\theta) \cdot \nabla\ell_i(\theta)$ versus $\nabla\ell_{\mathrm{val}}(\theta) \cdot \nabla\ell_j(\theta)$. However, at moderate learning rates, the second-order term becomes significant. For example, two datasets $D_i$ and $D_j$ that have better first-order gradient alignment can reverse their ordering once the quadratic terms outweigh this difference at moderate values of $\eta$. Overall, the ranking flip can occur as the learning rate moves beyond the tiny regime, where the higher-order curvature term can dominate the change in the loss, override first-order alignment, and ultimately flip the dataset rankings.

## 5 TRAINING SMALL PROXY MODELS WITH TINY LEARNING RATES IMPROVES TRANSFERABILITY TO TUNED TARGET MODELS

To reduce hyperparameter sensitivity in data recipe ablation and better align with the practical goal of identifying the recipe with the highest performance potential, one straightforward approach is to sweep through extensive hyperparameter configurations for each candidate data recipe and compare their optimal validation loss across all settings. However, this requires numerous training runs per candidate data recipe, which can be prohibitively expensive even with smaller proxy models. Moreover, training randomness can introduce significant variance in results, further reducing the reliability of such comparisons. Building on insights from our one-step gradient update analysis, we propose a simple yet effective solution: *train proxy models with very small learning rates* and rank data recipes by their validation losses under this regime.

## 5.1 EMPIRICAL FINDINGS & HIGH-LEVEL INTUITION

**Notations.** We denote the optimal validation loss achievable for a dataset $D$ by $\ell_{\text{val-opt}}(\theta(D)) = \min_{\eta \in \Lambda} \ell_{\text{val}}(\theta(D; \eta))$ where the minimum is taken over the full hyper-parameter space $\Lambda$. In our experiment, this optimum is approximated via a wide range of hyperparameter sweeps. Our pilot study (see Section 6 and Appendix D) shows that dataset rankings are *most* sensitive to the learning rate, while relatively stable across other major hyperparameters such as batch size. Therefore, throughout this section, we simplify notation and write $\theta(D; \eta)$ to denote a model trained with learning rate $\eta$ while all other hyperparameters are fixed to standard values from prior literature (details in Appendix D). Given two datasets $D_i$ and $D_j$, we denote by $\Delta \ell_{\text{val}}^{(i,j)}(\theta, \eta)$ the difference in validation loss when a model $\theta$ is trained on these datasets using a fixed learning rate $\eta$, and by $\Delta \ell_{\text{val-opt}}^{(i,j)}(\theta)$ the difference in their optimal validation losses after full hyperparameter tuning:

$$\Delta \ell_{\text{val}}^{(i,j)}(\theta, \eta) = \ell_{\text{val}}(\theta(D_i; \eta)) - \ell_{\text{val}}(\theta(D_j; \eta)), \quad \Delta \ell_{\text{val-opt}}^{(i,j)}(\theta) = \ell_{\text{val-opt}}(\theta(D_i)) - \ell_{\text{val-opt}}(\theta(D_j))$$

Our objective is to identify a specific learning rate $\eta_0$ for proxy models that reliably preserves the ranking of datasets based on their fully-optimized performance on large target models:

$$\text{sign}(\Delta \ell_{\text{val}}^{(i,j)}(\theta_{\text{proxy}}, \eta_0)) = \text{sign}(\Delta \ell_{\text{val-opt}}^{(i,j)}(\theta_{\text{tgt}})) \tag{1}$$

We present two key empirical findings (settings detailed in Appendix D) and the high-level intuition behind them, which motivates our solution of training proxy models with *tiny learning rates*.

**Finding I: For the same model, tiny learning rate performance correlates with optimal hyperparameter-tuned performance.** We first investigate the correlation between the losses for the same model architectures trained by different learning rates. Figure 4(a) shows the correlation between GPT2-Small's validation loss on the Pile validation split when trained with a tiny learning rate versus its minimum validation loss achieved through extensive hyperparameter sweeps, evaluated across 23 data recipes. The strong correlation demonstrates that for a given model architecture, training with a tiny learning rate provides a reliable proxy for optimized performance after full hyperparameter tuning. The relative utility of datasets, as measured by tiny learning rate training, is preserved when the same model is optimally configured. Formally, for small $\eta_0$, we observe that

$$\text{sign}(\Delta \ell_{\text{val}}^{(i,j)}(\theta_{\text{proxy}}, \eta_0)) = \text{sign}(\Delta \ell_{\text{val-opt}}^{(i,j)}(\theta_{\text{proxy}})) \tag{2}$$

for most dataset pairs $(D_i, D_j)$. **Intuition:** In one-step gradient update, when the learning rate $\eta \to 0$, model updates are dominated by the first-order gradient alignment term $\nabla \ell_{\text{val}}(\theta) \cdot \nabla \ell(\theta)$. This alignment measures the distributional similarity between training and validation datasets from the neural network's perspective, and is a metric widely used in data selection literature (Wang et al., 2024a; Fan et al., 2024b;a). Therefore, the validation loss at an infinitesimal learning rate effectively upper-bounds the irreducible train–validation distribution gap, and its ranking across datasets is strongly correlated with the best loss each dataset can achieve after full hyperparameter tuning.

**Finding II: Dataset rankings remain consistent across model scales when both use tiny learning rates.** In Figure 4(b), we visualize the heatmap of Spearman rank correlation coefficients between proxy and target model dataset rankings across various learning rate combinations. The lower-left region of the heatmap shows that when both proxy and target models are trained with tiny learning rates, the cross-scale transferability of dataset rankings achieves nearly perfect correlation. Formally, this suggests that when both $\eta_0$ and $\eta_1$ are small, we have

$$\text{sign}(\Delta \ell_{\text{val}}^{(i,j)}(\theta_{\text{proxy}}, \eta_0)) = \text{sign}(\Delta \ell_{\text{val}}^{(i,j)}(\theta_{\text{tgt}}, \eta_1)) \tag{3}$$

for most dataset pairs $(D_i, D_j)$. **Intuition:** As we discussed in the one-step gradient update analysis in Section 4, training with tiny learning rates minimizes the higher-order interactions that can confound data recipe comparison when using varying learning rates. While gradient alignment (the first-order term) is still model-dependent, its relative ranking across different candidate datasets tends to be preserved across model architectures (Sankararaman et al., 2020; Vyas et al., 2023). When both proxy and target models operate in the tiny-LR regime, their loss rankings are primarily determined by the distributional similarity between training and validation data. As a result, the validation-loss orderings remain relatively stable across different model scales, despite their differing capacities.

Collectively, the empirical observations in (2) and (3) together lead to (1) when using a small learning rate, motivating our strategy of ranking datasets by their tiny-LR losses on proxy models. Empirical evidence of this high-level intuition can be found in Appendix D.2.4.

## 5.2 THEORETICAL ANALYSIS

Section 5.1 uses a simplified one-step gradient update analysis to provide high-level intuition for the two empirical findings. To further support our findings, we formally prove that training random feature models aligns with the two empirical findings. We choose the random feature model because it is one of the simplest models whose training dynamics at different learning rates and scales can be approximately characterized mathematically. As the training dynamics of deep neural networks are usually non-tractable, a series of recent works have used random feature models as proxies to understand the dynamics of neural networks under model scaling (Bordelon et al., 2024; Lin et al., 2024; Medvedev et al., 2025). The random feature model is also closely related to the Neural Tangent Kernel (NTK), a key concept in the theory of deep neural networks that has often yielded valuable insights for developing data-centric techniques (Arora et al., 2019; Wang et al., 2024b).

**Theorem 1** (Informal). *Given two candidate data distributions $D_{\mathrm{A}}$ and $D_{\mathrm{B}}$, if the width of the random feature model is larger than a threshold, then after training on both datasets with learning rates $\eta$ small enough, with high probability, the relative ordering of $\ell_{\mathrm{val}}(\theta(D_{\mathrm{A}};\eta))$ and $\ell_{\mathrm{val}}(\theta(D_{\mathrm{B}};\eta))$ is the same as the relative ordering of the two validation loss values at the infinite-width limit.*

See Appendix B for complete theorem statements and proofs.

**Interpretation.** This theorem establishes that tiny-learning-rate training preserves dataset rankings relative to the *infinite-width optimum*, which serves as a theoretical anchor for understanding cross-scale transferability. The infinite-width model provides a theoretical reference point for understanding large target models after extensive hyperparameter tuning. In the infinite-width limit, training random feature models is equivalent to solving a kernel regression problem, and the obtained solution captures the optimal validation loss achievable within the function class. Our theory demonstrates that both optimally-tuned large target models and tiny-learning-rate proxy models converge to the same dataset rankings dictated by this infinite-width optimum as model width increases. Consequently, dataset rankings from small-scale training with low learning rates are consistent with those from optimally-tuned large-scale training with high probability.

**Intuition & Proof sketch.** Similar to our intuition from one-step gradient update, our analysis of random feature models decomposes the change in validation loss under SGD into a deterministic *drift* term that captures the true quality gap between datasets and a *variance* term from stochastic updates. When a sufficiently wide model is trained with a small learning rate, the drift dominates the variance, so the observed sign of the loss difference matches the sign of the infinite-width optimum gap with high probability. This formalizes the intuition that small learning rates suppress the curse of higher-order effects that otherwise scramble rankings.

## 5.3 AN EMPIRICAL CHARACTERIZATION OF "TINY" LEARNING RATES

Our analysis suggests the potential benefits of training proxy models with tiny learning rates. A natural question is *how small should the learning rate be for reliable proxy training runs?* We provide both theoretical guidance and practical recommendations, with extended discussion in Appendix C.

**Theoretical guidance.** Our theoretical analysis (detailed in Appendix C) establishes an upper bound for what constitutes a "tiny" learning rate in the context of one-step mini-batch SGD. At a high level, we show that $\eta_{\mathrm{tiny}}$ needs to be far smaller than $1/\lambda_{\max}$, where $\lambda_{\max}$ is the top eigenvalue of the validation loss Hessian. We stress that one-step analysis has often been shown to give quantitatively useful guidance when coupled with empirical validation (e.g. McCandlish et al. (2018); Smith et al. (2018)). Following the established precedent, we treat our derived bound as principled guidance rather than as a rigid constraint. In Appendix C, we use model checkpoints that have undergone short warmup training to estimate $\eta_{\mathrm{tiny}}$'s upper bound, and we show that these theoretically-predicted values consistently fall within the empirically-identified regime of learning rates with high transferability.

**Choosing $\eta_{\mathbf{tiny}}$ in practice: a simple rule of thumb.** While estimating $\eta_{\mathrm{tiny}}$ from proxy model checkpoints is computationally inexpensive, we find that a simpler heuristic suffices for most LLM pretraining scenarios. In Appendix C, we discuss the relationship between $\eta_{\mathrm{tiny}}$ and the standard learning rate for language model pretraining. Empirically, we find that *using learning rates 1-2 orders of magnitude smaller than standard choices of learning rates* generally works well. For language model pretraining, this typically translates to learning rates of approximately $10^{-5}$ to $10^{-6}$. In Section 6.2, we will see that these values work consistently well across various model scales.

Practical implementation also requires consideration of the lower bounds of $\eta_{\text{tiny}}$ due to numerical precision. However, we find that the recommended range of $10^{-5}$ to $10^{-6}$ lies comfortably above the threshold where numerical errors become significant in standard training. Our experiments confirm that learning rates in this range maintain numerical stability while achieving the desired transferability properties. We provide additional analysis of numerical considerations in Appendix C.

# 6 EXPERIMENTS

## 6.1 EXPERIMENT SETTINGS

**Evaluation protocol.** We conduct comprehensive evaluations across multiple model architectures (GPT2, Pythia, OPT) spanning sizes from 70M to 1B parameters. We assess transferability using both the validation loss across diverse domains from The Pile corpus (Gao et al., 2020) and downstream benchmark performance on HellaSwag, Winogrande, OpenBookQA, ARC-Easy, and CommonsenseQA. Following standard practices, we primarily set the token-per-parameter ratio to 20, adhering to Chinchilla's compute-optimal ratio (Hoffmann et al., 2022). Additional ablation studies examining overtraining scenarios (with ratios up to 160) are presented in Appendix D.2.2. All experiments use single-epoch training without sample repetition, following standard LLM pretraining conventions. Crucially, target models undergo *dataset-specific hyperparameter optimization* to align with our proposed methodology, with tuning budget held constant across all experiments.

**Data Recipes.** To comprehensively evaluate proxy model transferability across the diverse data curation decisions faced in production scenarios, we construct 23 data recipes spanning four critical dimensions of language data curation. **(1) Domain composition and ablation.** We create 10 domain mixture variations from The Pile (Gao et al., 2020). Specifically, 4 recipes maintain a fixed 70% Pile-CC allocation while varying the remaining 30% between StackExchange and ArXiv. Additionally, 6 recipes are created by excluding one domain from the full Pile dataset to assess domain importance. **(2) Existing dataset comparison.** We evaluate 3 widely-adopted pretraining corpora: C4 (Raffel et al., 2020), DCLM-baseline (Li et al., 2024), and RefinedWeb (Penedo et al., 2023). They all originate from Common Crawl but vary in curation strategies such as data filtering criteria and deduplication algorithms. **(3) Scoring-based data filter.** We construct 6 data recipes using different mixing ratios of head-middle versus tail partitions from RedPajama-V2 Weber et al. (2024), ranging from 50:50 to 100:0. These partitions are defined by the perplexity scores from a 5-gram Kneser-Ney model trained on Wikipedia (Wenzek et al., 2020). **(4) Deduplication.** We create 4 variants of DCLM-baseline by modifying the stringency of Massive Web Repetition Filters (Rae et al., 2021), adjusting n-gram and line/paragraph repetition thresholds. The detailed description for the data recipes considered in this work is summarized in Appendix D.1.4 and Table 1.

**Evaluation Metrics.** We report two complementary quantities that capture (i) rank fidelity and (ii) practical decision risk. **Spearman rank correlation.** This measures the monotone agreement between the proxy-induced ranking and the target-scale ranking, where the target ranking is obtained by training the target model on each dataset with its *own tuned hyperparameters*. **Top-$k$ Decision Regret.** This addresses the practical question: if a data team selects the top-$k$ data recipes ranked by the proxy model for full-scale evaluation, how suboptimal might their final choice be compared with the best data recipe? Formally, we compute the performance difference between the truly optimal dataset and the best-performing dataset among the proxy's top-$k$ selections, with both datasets evaluated using their individually optimized hyperparameters on the target model.

Due to space constraints, we leave detailed model training and other settings to Appendix D.1.

**Remark.** *A recently released benchmark suite, DataDecide (Magnusson et al., 2025), provides up to 1B pre-trained models for efficient evaluation of proxy model transferability. However, they use a fixed training configuration across all datasets, which may cause the conclusions to overfit to the specific hyperparameter configurations, as we discussed earlier. Therefore, we conduct our own evaluation in which all target models are hyperparameter-tuned for each data recipe.*

**Remark** (Computational challenges in evaluating proxy model transferability). *We consider large target models of size up to 1B parameters, which matches the largest target model size in the established DataDecide benchmark suite (Magnusson et al., 2025) developed by AI2 for evaluating proxy model transferability. Our experimental protocol requires training target models with extensive dataset-specific hyperparameter optimization, which further amplifies computational requirements in*

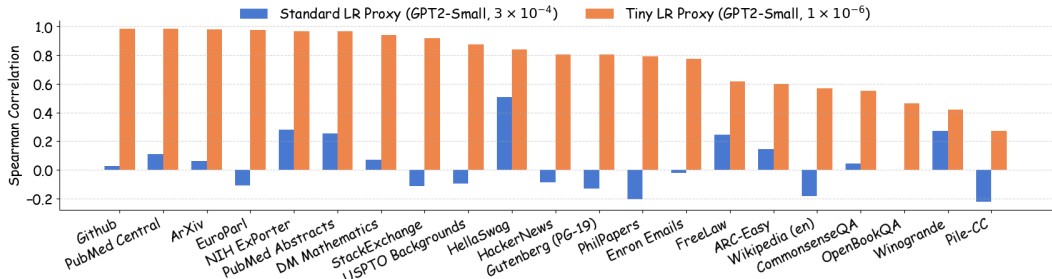

Figure 5: Average rank correlation between proxy (GPT2-Small) and target (GPT2-Large) for the loss computed over a variety of validation domains (from Pile) and downstream benchmarks. Compared to the standard learning rate used in the literature (Karpathy, 2022), a tiny learning rate consistently improves the proxy model's rank transferability, often by a significant margin.

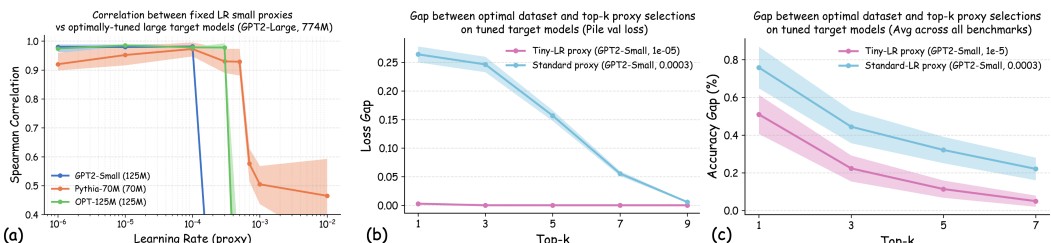

Figure 6: Dataset ranking consistency between proxy and target models across different learning rates, evaluated on validation loss and downstream benchmarks. **(a)** Spearman rank correlation between dataset rankings from proxy models (GPT2-Small, Pythia-70M, OPT-125M) and larger target model (GPT2-Large) as a function of proxy model learning rate, evaluated on aggregated Pile validation loss. **(b)** Top-$k$ gap between optimal dataset and proxy selections on hyperparameter-optimized target models, measured using Pile validation loss. **(c)** Top-$k$ gap measured using the averaged accuracies across the 5 benchmarks we considered (see Section 6.1). Gaps approaching zero indicate that proxy models reliably identify datasets that perform optimally on larger models. Shaded areas show 95% bootstrap CIs computed by resampling seeds (3 runs per dataset). Additional results are available in Appendix D.2.1.

*evaluation. However, this is essential for properly evaluating each dataset's optimal performance potential as advocated in our refined objective. Throughout this project, we completed $> 20,000$ model training runs (including both small proxy models and large target models). We leave the exploration of efficient evaluation protocols for proxy-to-target transferability as a future direction.*

## 6.2 RESULTS

As shown in Figure 6 (a), training proxy models with standard learning rates ($3 \times 10^{-4}$ for GPT2-Small (Karpathy, 2022) and OPT-125M (Zhang et al., 2022), $10^{-3}$ for Pythia-70M (Biderman et al., 2023)) yields only modest ranking agreement, with Spearman rank correlation $\rho < 0.75$. This poor rank consistency can lead to suboptimal data selection decisions and performance degradation in large-scale model training. However, when the learning rate drops below $1 \times 10^{-4}$, we observe substantial improvements in ranking agreement. Under these smaller learning rates, the proxy models achieve performance ranking correlations above 0.92 across all three architectures. Figure 5 extends these results beyond aggregated validation loss to examine how small learning rates affect the reliability of proxy models across individual Pile domains and downstream benchmarks. The results demonstrate that training proxy models with reduced learning rates improves data recipe ranking correlations across nearly all evaluation metrics. In contrast, standard learning rate proxies exhibit catastrophic failure across most metrics, yielding near-zero or negative correlations in many domains.

Figure 6 (b) and (c) demonstrate the practical implications of these ranking correlations through the top-$k$ decision regret metric. When data teams select the top-$k$ data recipes for large-scale model training based on small-scale experiments, proxies trained with tiny learning rates consistently

minimize the performance gap relative to the ground-truth optimal data recipes for the target model. In contrast, standard learning rate proxies can incur $> 0.25$ validation loss degradation when $k$ is small. This stark difference demonstrates that conducting small-scale ablation studies with smaller learning rates can significantly enhance the reliability of data decisions in large-scale model training.

# 7 CONCLUSION & FUTURE WORKS

We demonstrate that standard data recipe ablations are brittle in small-scale settings; even slight adjustments to the learning rate can yield inconsistent findings. To address this, we proposed a simple yet effective solution: using *tiny* learning rates during proxy model training. In the following, we discuss the scope of this work and outline several potential future directions.

**Extending proxy model techniques to multi-epoch and curriculum learning.** This work focuses on single-epoch LLM pretraining. Designing effective proxy training runs for multi-epoch scenarios presents significant challenges. A key question is how to subsample the full dataset while properly accounting for the repetition patterns that emerge across epochs. Similar technical difficulties arise in curriculum learning. Given the growing data scarcity, developing principled small-scale proxy training protocols for multi-epoch and curriculum-based learning is an important future direction.

**Joint optimization of data and training configurations.** As we mentioned in the Introduction, our approach serves as a simple remedy to the proxy-model-based approach. However, because the challenge stems from the strong coupling between training configurations and datasets, we argue that, in the long term, datasets and tunable training configurations should be jointly optimized. Recent advances in gradient-based hyperparameter optimization (Lorraine et al., 2020) and algorithm unrolling (Chen et al., 2022) may provide inspiration and building blocks for this direction.

ACKNOWLEDGEMENT

This work is supported in part by the Apple PhD Fellowship. Ruoxi Jia and the ReDS lab acknowledge support through grants from the National Science Foundation under grants IIS-2312794, IIS-2313130, and OAC-2239622.

We thank Lin Chen, Mohammadhossein Bateni, Jennifer Brennan, Clayton Sanford, and Vahab Mirrokni at Google Research for their helpful feedback on the preliminary version of this work.

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

## A EXTENDED BACKGROUND & RELATED WORKS

### A.1 TAXONOMY OF PROXY-MODEL-BASED METHODOLOGY

In the literature, proxy-model-based approaches for data quality optimization typically fall into three main categories, each of which uses different strategies to reduce computational costs.

**(1) Model-scale proxies** maintain the full training schedule but use models with fewer parameters (e.g., reduced layer count or hidden dimensions) for the proxy training run. This approach is relatively common in early literature where the dataset sizes are relatively small (Coleman et al., 2020). We found one example in large language model pretraining (Xie et al., 2023).

**(2) Data-scale proxies** preserve the target model architecture while drastically reducing the number of training iterations or tokens processed (Kang et al., 2024). This strategy is commonly used for optimizing training data in continual pretraining scenarios and is particularly prevalent in developing compact models such as Phi-4 (Abdin et al., 2024). The approach operates under the fundamental assumption that early training dynamics provide sufficient predictive signals to forecast long-term convergence properties and final model performance accurately.

**(3) Ratio-preserving proxies** simultaneously reduce both model and data scale, generally maintaining consistent scaling relationships between them (e.g., (Li et al., 2024; Magnusson et al., 2025)). In language model training, practitioners typically preserve the parameter-to-token ratio prescribed by Chinchilla scaling laws (Hoffmann et al., 2022). This balanced approach during downscaling ensures that proxy models operate in appropriate regimes—neither underfitting nor overfitting relative to their capacity—which better mimics the learning dynamics of full-scale target models.

**Scope of this work:** Our work primarily focuses on **(3) Ratio-preserving proxies**, representing the most widely adopted approach in large-scale language model development. A systematic study examining the other two proxy training methodologies and their comparative effectiveness remains an important direction for future research.

**Remark** (Architecture and scale considerations.). *We emphasize that this work focuses specifically on optimizing training hyperparameters for proxy models, particularly the learning rate, rather than addressing questions about proxy model architecture or scale selection. While our experiments demonstrate the effectiveness of tiny learning rates across three different model families (GPT-2, Pythia, and OPT) with varying model sizes, we do not delve into how architectural choices or the proxy-to-target size ratio affect the reliability of dataset selection. The question of how large a proxy model should be relative to the target model, and whether certain architectural families provide better transferability than others, is an important future work. Our theoretical analysis in Section 5.2 for random feature models provides initial insights into width requirements, but extending this understanding to modern transformer architectures and establishing principled guidelines for proxy model sizing remains an open challenge.*

### A.2 EXTENDED RELATED WORKS

**Proxy models in data-centric machine learning.** Proxy models have emerged as a standard efficiency solution across the literature of data-centric machine learning algorithms that require model retraining. This approach originated from early works demonstrating that feature representations from small, computationally efficient proxy models could effectively substitute for representations from larger, more accurate target models in the task of coreset selection and active learning (Lewis & Catlett, 1994; Coleman et al., 2020). The research community has leveraged this technique to accelerate various dataset optimization algorithms, including data mixture optimization (e.g., (Xie et al., 2023; Chen et al., 2023)), data point filtering (e.g., (Mekala et al., 2024; Ni et al., 2024)), active learning (e.g., (Wen et al., 2024)), and prompt engineering (e.g., (Liu et al., 2024a; Hong et al., 2024)). Recent works have even extended this paradigm to knowledge distillation (Rawat et al., 2024), where small proxy models guide the early training stages of larger models.

**Data-dependent scaling laws.** A parallel research direction explores using proxy model performance to directly predict target model performance by fitting scaling laws. For instance, (Anugraha et al., 2024) fits regression models on proxy performance metrics to forecast target model outcomes. Brandfonbrener et al. (2024) shows that test losses between two models trained on separate datasets can follow predictable scaling patterns, though their work specifically examines paired model and

dataset size configurations. The line of works that are most relevant to this paper (Liu et al., 2024b; Ye et al., 2024) use small proxy models to fit data mixture-dependent scaling laws that predict target model performance across different mixture configurations. They then use these predictions to optimize the composition of training data mixtures. Another work by (Kang et al., 2024) explores an alternative proxy approach, where models with identical architecture are trained for fewer iterations, and the resulting performance curves are extrapolated through scaling laws to predict outcomes at extended training durations. However, these approaches share a limitation: all of the proxy training runs use identical, fixed hyperparameters across all data mixtures. As discussed in our paper, this approach can inadvertently favor datasets that align well with the chosen training configuration, potentially distorting the resulting data mixture recommendations. Revisiting these data mixture optimization methods with hyperparameter-tuned proxy models is an interesting and promising way to extend our findings.

**Understanding proxy-to-target transferability.** Despite the widespread use of proxy models, relatively few studies have systematically investigated the reliability and conditions under which proxy model conclusions transfer effectively to large target models. (Khaddaj et al., 2025) empirically showed that proxy model losses generally correlate strongly with those of large-scale models across various training datasets, though the relationship depends somewhat on specific test domains or downstream tasks. Their work demonstrates that data attribution and selection tasks can be reliably performed using smaller proxy models. More recently, (Magnusson et al., 2025) released DataDecide, a comprehensive suite containing over 30,000 checkpoints spanning 25 corpora. Their analysis shows that scaling law-based approaches do not consistently outperform single-scale proxy models for performance prediction. Theoretical analysis in this area remains notably sparse, with (Gu et al., 2025) providing a rare exception by proving that the optimal ratio between factual and web-scraped data can shift significantly when scaling from small proxy models to larger target models. However, these studies neither identified the specific conditions that lead to high transferability nor examined how training hyperparameters affect proxy model reliability. Our work significantly extends this line by demonstrating that proxy training runs with *tiny learning rates* lead to strong transferability across model scales, offering a computationally efficient yet more reliable enhancement to existing proxy-model pipelines.

**Connections to hyperparameter transferability.** The proxy-model-based dataset selection problem bears striking similarities to well-documented challenges in hyperparameter optimization, where configurations optimized for small-scale models often transfer poorly to larger architectures (Li et al., 2020; Yang et al., 2022). While solutions like $\mu$-parameterization have been proposed to address hyperparameter transferability across model scales, their application to dataset transferability remains unexplored. $\mu$-parameterization might theoretically improve data transferability when small and large models train for identical iteration counts, but proxy-based techniques typically train on substantially reduced time. An interesting future direction is to explore whether analogous principles can be developed specifically for training data's transferability. On the other hand, our work demonstrates that simply training proxy models with tiny learning rates, without modifying any other training configurations, significantly improves cross-scale data transferability.

# B  THEORETICAL ANALYSIS

In this section, we analyze two-layer random feature networks and prove that the transferability to datasets' minimum achievable performance holds when the proxy models are trained with tiny learning rates.

## B.1  SETUP

**Random Feature Networks.** We analyze two-layer random feature networks, also known as random feature models. Let $\sigma : \mathbb{R} \to \mathbb{R}$ be a Lipschitz function. The weights in the first layer $\boldsymbol{U} := (\boldsymbol{u}_1, \ldots, \boldsymbol{u}_m) \in \mathbb{R}^{d \times m}$ are drawn independently from a bounded distribution $\mathcal{U}$ on $\mathbb{R}^d$. For all $\boldsymbol{x}$ in a compact input space $\mathcal{X} \subseteq \mathbb{R}^d$, define the feature map

$$\phi_i^{(\boldsymbol{U})}(\boldsymbol{x}) = \sigma(\langle \boldsymbol{u}_i, \boldsymbol{x} \rangle), \qquad \phi^{(\boldsymbol{U})}(\boldsymbol{x}) = (\phi_1^{(\boldsymbol{U})}(\boldsymbol{x}), \ldots, \phi_m^{(\boldsymbol{U})}(\boldsymbol{x}))^\top \in \mathbb{R}^m. \tag{4}$$

The second layer is parameterized by a weight vector $\boldsymbol{\theta} \in \mathbb{R}^m$. Given the first-layer weights $\boldsymbol{U}$ and the second-layer weights $\boldsymbol{\theta}$, the two-layer network outputs

$$f_{\boldsymbol{\theta}}^{(\boldsymbol{U})}(\boldsymbol{x}) = \frac{1}{\sqrt{m}} \langle \boldsymbol{\theta}, \phi^{(\boldsymbol{U})}(\boldsymbol{x}) \rangle. \tag{5}$$

This function computes a linear combination of randomly generated nonlinear features of the input $\boldsymbol{x}$.

**Data Distribution.** A data distribution $D$ is a probability distribution over $\mathcal{X} \times \mathbb{R}$, where the marginal distribution on inputs $\boldsymbol{x}$, denoted as $D_{\mathrm{x}}$, is assumed to admit a continuous density function over $\mathcal{X}$. Each sample $(\boldsymbol{x}, y) \sim D$ consists of an input $\boldsymbol{x}$ and a label $y$. We say that $D$ has *no label noise* if there exists a target function $f_D^* : \mathcal{X} \to \mathbb{R}$ such that $y = f_D^*(\boldsymbol{x})$ almost surely for $(\boldsymbol{x}, y) \sim D$.

**Training.** We define $\mathcal{P}_m(D; \eta, B, T)$ as the distribution of $(\boldsymbol{\theta}, \boldsymbol{U})$ of a width-$m$ network trained on a data distribution $D$ with the following procedure. Define the MSE loss as:

$$\ell(f; \boldsymbol{x}, y) := \frac{1}{2}(f(\boldsymbol{x}) - y)^2, \qquad \mathcal{L}_D(\boldsymbol{\theta}; \boldsymbol{U}) = \mathbb{E}_{(\boldsymbol{x}, y) \sim D}\left[\ell(f_{\boldsymbol{\theta}}^{(\boldsymbol{U})}; \boldsymbol{x}, y)\right]. \tag{6}$$

We randomly sample the first-layer weights $\boldsymbol{U} \sim \mathcal{U}^m$ as described above and freeze them. We then optimize the second-layer weights $\boldsymbol{\theta}$ using one-pass mini-batch SGD with batch size $B$ and learning rate $\eta$, starting from $\boldsymbol{\theta}_0 = \boldsymbol{0}$. The update rule is given by:

$$\boldsymbol{\theta}_{t+1} = \boldsymbol{\theta}_t - \frac{\eta}{B} \sum_{b=1}^{B} \frac{\partial}{\partial \boldsymbol{\theta}} \left(\ell(f_{\boldsymbol{\theta}}^{(\boldsymbol{U})}; \boldsymbol{x}_{t,b}, y_{t,b})\right)\bigg|_{\boldsymbol{\theta} = \boldsymbol{\theta}_t}, \tag{7}$$

where $(\boldsymbol{x}_{t,b}, y_{t,b})$ is the $b$-th training input and label in the $t$-th batch, sampled independently from $D$. $\mathcal{P}^{(m)}(D; \eta, B, T)$ is the distribution of $(\boldsymbol{\theta}_T, \boldsymbol{U})$ obtained by running the above procedure for $T$ steps.

**Validation.** We fix a data distribution $D_{\mathrm{val}}$ for validation, called the validation distribution. The validation loss is given by

$$\mathcal{L}_{\mathrm{val}}(f) = \mathbb{E}_{(\boldsymbol{x}, y) \sim D_{\mathrm{val}}}\left[\ell(f; \boldsymbol{x}, y)\right]. \tag{8}$$

Given a training distribution $D$ with no label noise, we define the expected validation loss of a width-$m$ network trained on $D$ with batch size $B$ and learning rate $\eta$ for $T$ steps as

$$\mathcal{I}_{\mathrm{val}}^{(m)}(D; \eta, B, T) = \mathbb{E}_{(\boldsymbol{\theta}, \boldsymbol{U}) \sim \mathcal{P}^{(m)}(D; \eta, B, T)}\left[\mathcal{L}_{\mathrm{val}}(f_{\boldsymbol{\theta}}^{(\boldsymbol{U})})\right]. \tag{9}$$

Further, we define the best achievable validation loss of a width-$m$ network trained on $D$ with batch size $B$ for $T$ steps (after tuning the learning rate $\eta$) as

$$\mathcal{I}_{\mathrm{val\text{-}opt}}^{(m)}(D; B, T) = \min_{\eta > 0} \left\{\mathcal{I}_{\mathrm{val}}^{(m)}(D; \eta, B, T)\right\}, \tag{10}$$

$$\eta_{\mathrm{opt}}^{(m)}(D; B, T) = \operatorname*{argmin}_{\eta > 0} \left\{\mathcal{I}_{\mathrm{val}}^{(m)}(D; \eta, B, T)\right\}. \tag{11}$$

**Understanding the Infinite-Width Limit.** Before stating our assumptions on the data distributions, we first provide some intuition on the infinite-width limit of the random feature model. A random feature model with width $m$ naturally induces a kernel function $K^{(\boldsymbol{U})}(\boldsymbol{x}, \boldsymbol{x}') = \frac{1}{m} \langle \phi^{(\boldsymbol{U})}(\boldsymbol{x}), \phi^{(\boldsymbol{U})}(\boldsymbol{x}') \rangle$

on the input space $\mathcal{X}$. As $m \to \infty$, the randomness in the features $\phi^{(U)}(\boldsymbol{x})$ averages out due to the law of large numbers and the kernel $K^{(U)}(\boldsymbol{x}, \boldsymbol{x}')$ converges to a deterministic kernel $K(\boldsymbol{x}, \boldsymbol{x}')$:

$$K(\boldsymbol{x}, \boldsymbol{x}') = \mathbb{E}_{\boldsymbol{u} \sim \mathcal{U}}\left[\sigma(\langle \boldsymbol{u}, \boldsymbol{x} \rangle)\sigma(\langle \boldsymbol{u}, \boldsymbol{x}' \rangle)\right]. \tag{12}$$

In this limit, the kernel $K$ captures the expressive power of the infinite-width model: learning in this regime becomes equivalent to kernel regression using $K$. The function class defined by the model becomes a Reproducing Kernel Hilbert Space (RKHS) associated with $K$, denoted as $\mathcal{H}_K$. The properties of this kernel—especially whether it is positive definite and how well it aligns with the target function—directly influence the learnability of the problem. With this in mind, we now formalize the assumptions we make about the data distributions in our theoretical analysis.

**Well-Behaved Distributions.** We say that a distribution $D$ with no label noise is well-behaved if the following conditions are satisfied:

1. **Full Support on the Input Space.** The probability density of $D_{\mathrm{x}}$ is strictly positive on the input space $\mathcal{X}$. This ensures that $D_{\mathrm{x}}$ has full support on $\mathcal{X}$.

2. **Positive-Definite Kernel.** The kernel $K(\boldsymbol{x}, \boldsymbol{x}')$ has strictly positive minimum eigenvalue on $L^2(D_{\mathrm{x}})$, the square-integrable function space under $D_{\mathrm{x}}$. This prevents the kernel function from being degenerate.

3. **Realizability at Infinite Width.** The target function $f_D^*$ lies in the RKHS $\mathcal{H}_K$ associated with the kernel $K$. More specifically, there exists $\nu : \mathrm{supp}(\mathcal{U}) \to \mathbb{R}$ such that $\sup_{\boldsymbol{u} \in \mathrm{supp}(\mathcal{U})}|\nu(\boldsymbol{u})| < \infty$ and it holds for all $\boldsymbol{x} \in \mathcal{X}$ that

$$f_D^*(\boldsymbol{x}) = \mathbb{E}_{\boldsymbol{u} \sim \mathcal{U}}[\sigma(\langle \boldsymbol{u}, \boldsymbol{x} \rangle)\nu(\boldsymbol{u})] \tag{13}$$

   This condition ensures that $f_D^*$ can be learned by kernel regression with kernel $K$ and the random feature model with infinite width.

4. **Compatible with the Validation Distribution.** The distribution $D$ satisfies the following:

$$\lim_{m \to \infty} \lim_{T \to \infty} \mathbb{E}_{(\boldsymbol{\theta}; \boldsymbol{U}) \sim \mathcal{P}_m^*(D; B, T)}\left[\mathcal{L}_D(\boldsymbol{\theta}; \boldsymbol{U})\right] = 0, \tag{14}$$

   where $\mathcal{P}_m^*(D; B, T) = \mathcal{P}_m(D; \eta_{\mathrm{opt}}^{(m)}(D; B, T), B, T)$ is the distribution of $(\boldsymbol{\theta}, \boldsymbol{U})$ obtained by running SGD when the learning rate is optimally tuned for minimizing the validation loss. Intuitively, this condition means that asymptotically, the best way to minimize the validation loss is to fit $f_D^*$ exactly. In other words, although there is a distribution shift between $D$ and $D_{\mathrm{val}}$, when the network is sufficiently wide and the training is sufficiently long, tuning the learning rate $\eta$ for minimizing the validation loss is essentially equivalent to tuning it for fitting $f_D^*$ better on $D$. This is consistent with the typical phenomenon of validation loss, which usually decreases as training progresses. It is easy to see that this condition implies the following:

$$\lim_{m \to \infty} \lim_{T \to \infty} \mathcal{I}_{\mathrm{val\text{-}opt}}^{(m)}(D; B, T) = \mathcal{L}_{\mathrm{val}}(f_D^*), \tag{15}$$

   where we call $\mathcal{L}_{\mathrm{val}}(f_D^*)$ the *best achievable validation loss* of the training distribution $D$.

**Main Result.** Now we are ready to state our main result. Suppose we have two training distributions $D_{\mathrm{A}}$ and $D_{\mathrm{B}}$. We want to know training on which distribution leads to a smaller validation loss. Mathematically, we want to know which one has a smaller $\mathcal{I}_{\mathrm{val\text{-}opt}}^{(m)}(\,\cdot\,; B, T)$. The following theorem shows that comparing $\mathcal{I}_{\mathrm{val}}^{(m)}(\,\cdot\,; \eta, B, T)$ with a sufficiently small learning rate $\eta$ reliably preserves this ordering.

**Theorem 2** (Main Theorem). *Let $D_{\mathrm{A}}$ and $D_{\mathrm{B}}$ be two* well-behaved *training distributions with no label noise. Assume that the best achievable validation loss of $D_{\mathrm{A}}$ is different from that of $D_{\mathrm{B}}$, i.e.,*

$$\Delta_{\mathrm{AB}} := \mathcal{L}_{\mathrm{val}}(f_{D_{\mathrm{A}}}^*) - \mathcal{L}_{\mathrm{val}}(f_{D_{\mathrm{B}}}^*) \neq 0. \tag{16}$$

*Then, there exist constants $m_0, T_0, c_0, \alpha, \eta_{\max} > 0$ such that for all $m \geq m_0$, $T \geq T_0$, $B \geq 1$, if $\eta$ lies in the range $\frac{c_0}{T} \leq \eta \leq \min\{\alpha B m, \eta_{\max}\}$, then*

$$\mathrm{sign}\left(\mathcal{I}_{\mathrm{val}}^{(m)}(D_{\mathrm{A}}; \eta, B, T) - \mathcal{I}_{\mathrm{val}}^{(m)}(D_{\mathrm{B}}; \eta, B, T)\right) = \mathrm{sign}(\Delta_{\mathrm{AB}}). \tag{17}$$

**Interpretation.** The theorem establishes that the sign of the difference in the expected validation loss of two datasets, $\text{sign}(\mathcal{I}_{\text{val}}^{(m)}(D_{\text{A}}; \eta, B, T) - \mathcal{I}_{\text{val}}^{(m)}(D_{\text{B}}; \eta, B, T))$, when trained with a small, fixed learning rate $\eta$, is identical to the sign of the difference in their best achievable validation losses in the infinite-width limit, $\text{sign}(\Delta_{\text{AB}})$. For random feature models, this infinite-width limit, $\mathcal{L}_{\text{val}}(f_D^*)$, represents the theoretical optimum that a model can achieve after full hyperparameter tuning and sufficient scaling. Theorem 2 shows that for a given set of candidate data recipes, the ranking observed in the tiny-learning-rate regime accurately predicts the ranking according to the datasets' true potential. Importantly, as model size increases, the dataset rankings for optimally-tuned larger target models also converge to this infinite-width limit, thereby providing the necessary theoretical link to justify using small models with tiny learning rates for reliable data selection.

## B.2 PROOFS

To prove the main theorem, it suffices to show the following lemma.

**Lemma 3.** *Let $D$ be a training distribution that has no label noise and is compatible with $D_{\text{val}}$. Given $\Delta > 0$, there exist constants $m_0, T_0, c_0, \alpha, \eta_{\max} > 0$ such that for all $m \geq m_0$, $T \geq T_0$, $B \geq 1$, if $\eta$ lies in the range $\frac{c_0}{T} \leq \eta \leq \min\{\alpha B m, \eta_{\max}\}$, then*

$$\left| \mathcal{I}_{\text{val}}^{(m)}(D; \eta, B, T) - \mathcal{L}_{\text{val}}(f_D^*) \right| \leq \Delta. \tag{18}$$

*Proof of Theorem 2.* Applying Lemma 3 to $\mathcal{D} = D_{\text{A}}$ and $\mathcal{D} = D_{\text{B}}$ with $\Delta = \frac{\Delta_{\text{AB}}}{3}$, we have constants $m_0^{\text{A}}, T_0^{\text{A}}, c_0^{\text{A}}, \alpha^{\text{A}}, \eta_{\max}^{\text{A}} > 0$ and $m_0^{\text{B}}, T_0^{\text{B}}, c_0^{\text{B}}, \alpha^{\text{B}}, \eta_{\max}^{\text{B}} > 0$ such that for all $m \geq \max\{m_0^{\text{A}}, m_0^{\text{B}}\}$, $T \geq \max\{T_0^{\text{A}}, T_0^{\text{B}}\}$, $B \geq 1$, if $\eta$ lies in the range $\frac{\min\{c_0^{\text{A}}, c_0^{\text{B}}\}}{T} \leq \eta \leq \min\{\alpha^{\text{A}} B m, \alpha^{\text{B}} B m, \eta_{\max}^{\text{A}}, \eta_{\max}^{\text{B}}\}$, then

$$\left| \mathcal{I}_{\text{val}}^{(m)}(D_{\text{A}}; \eta, B, T) - \mathcal{L}_{\text{val}}(f_{D_{\text{A}}}^*) \right| \leq \frac{\Delta_{\text{AB}}}{3}, \quad \left| \mathcal{I}_{\text{val}}^{(m)}(D_{\text{B}}; \eta, B, T) - \mathcal{L}_{\text{val}}(f_{D_{\text{B}}}^*) \right| \leq \frac{\Delta_{\text{AB}}}{3}. \tag{19}$$

Therefore, $\text{sign}\left( \mathcal{I}_{\text{val}}^{(m)}(D_{\text{A}}; \eta, B, T) - \mathcal{I}_{\text{val}}^{(m)}(D_{\text{B}}; \eta, B, T) \right) = \text{sign}(\Delta_{\text{AB}})$ as desired. $\qquad\square$

In the rest of this section, we prove Lemma 3 for a fixed training distribution $D$. In the following, we define the following quantities depending on the first-layer weights $\boldsymbol{U}$:

$$\boldsymbol{H} := \frac{1}{m} \mathbb{E}_{(\boldsymbol{x}, y) \sim \mathcal{D}} \left[ \phi^{(\boldsymbol{U})}(\boldsymbol{x}) \phi^{(\boldsymbol{U})}(\boldsymbol{x})^\top \mid \boldsymbol{U} \right], \tag{20}$$

$$\boldsymbol{\beta} := \frac{1}{\sqrt{m}} \mathbb{E}_{(\boldsymbol{x}, y) \sim \mathcal{D}} \left[ y \phi^{(\boldsymbol{U})}(\boldsymbol{x}) \mid \boldsymbol{U} \right], \tag{21}$$

$$\boldsymbol{\mu} := \boldsymbol{H}^+ \boldsymbol{\beta} \in \underset{\boldsymbol{\theta} \in \mathbb{R}^m}{\arg\min} \, \mathcal{L}_D(\boldsymbol{\theta}; \boldsymbol{U}), \tag{22}$$

$$\boldsymbol{\Sigma} := \frac{1}{m} \mathbb{E}_{(\boldsymbol{x}, y) \sim \mathcal{D}} \left[ \left( y - f_{\boldsymbol{\mu}}^{(\boldsymbol{U})}(\boldsymbol{x}) \right)^2 \phi^{(\boldsymbol{U})}(\boldsymbol{x}) \phi^{(\boldsymbol{U})}(\boldsymbol{x})^\top \mid \boldsymbol{U} \right]. \tag{23}$$

### B.2.1 LOSS DECOMPOSITION

First, we present the following lemma that decomposes the validation loss of any random feature model into the best achievable validation loss plus some error terms.

**Lemma 4.** *For all $\boldsymbol{U} \in \mathbb{R}^{d \times m}$ and $\boldsymbol{\theta} \in \mathbb{R}^m$,*

$$\forall c > 0: \quad \mathcal{L}_{\text{val}}(f_{\boldsymbol{\theta}}^{(\boldsymbol{U})}) \leq (1 + c)\mathcal{L}_{\text{val}}(f_D^*) + \beta \cdot (1 + \tfrac{1}{c})\mathcal{L}_D(\boldsymbol{\theta}; \boldsymbol{U}), \tag{24}$$

$$\forall c \in (0, 1]: \quad \mathcal{L}_{\text{val}}(f_{\boldsymbol{\theta}}^{(\boldsymbol{U})}) \geq (1 - c)\mathcal{L}_{\text{val}}(f_D^*) - \beta \cdot (\tfrac{1}{c} - 1)\mathcal{L}_D(\boldsymbol{\theta}; \boldsymbol{U}), \tag{25}$$

*where $\beta := \sup_{\boldsymbol{x} \in \mathcal{X}} \frac{D_{\text{valx}}(\boldsymbol{x})}{D_{\text{x}}(\boldsymbol{x})}$.*

*Proof.* For the first inequality, since $(a + b)^2 \le (1 + c)a^2 + (1 + \frac{1}{c})b^2$ for all $c > 0$, we have

$$\mathcal{L}_{\text{val}}(f_{\boldsymbol{\theta}}^{(\boldsymbol{U})}) = \frac{1}{2}\mathbb{E}_{(\boldsymbol{x},y)\sim D_{\text{val}}}\left[(f_{\boldsymbol{\theta}}^{(\boldsymbol{U})}(\boldsymbol{x}) - y)^2\right]$$

$$\le \frac{1+c}{2}\mathbb{E}_{(\boldsymbol{x},y)\sim D_{\text{val}}}\left[(y - f_D^*(\boldsymbol{x}))^2\right] + \frac{1+\frac{1}{c}}{2}\mathbb{E}_{(\boldsymbol{x},y)\sim D_{\text{val}}}\left[(f_{\boldsymbol{\theta}}^{(\boldsymbol{U})}(\boldsymbol{x}) - f_D^*(\boldsymbol{x}))^2\right]$$

$$\le (1+c)\mathcal{L}_{\text{val}}(f_D^*) + \beta \cdot (1 + \tfrac{1}{c})\mathcal{L}_D(\boldsymbol{\theta}; \boldsymbol{U}),$$

where the last inequality follows from the fact that $\mathbb{E}_{(\boldsymbol{x},y)\sim D_{\text{val}}}\left[(f_{\boldsymbol{\theta}}^{(\boldsymbol{U})}(\boldsymbol{x}) - f_D^*(\boldsymbol{x}))^2\right] = \int_{\mathcal{X}}(f_{\boldsymbol{\theta}}^{(\boldsymbol{U})}(\boldsymbol{x}) - f_D^*(\boldsymbol{x}))^2 D_{\text{val}\boldsymbol{x}}(\boldsymbol{x})d\boldsymbol{x} \le \beta\int_{\mathcal{X}}(f_{\boldsymbol{\theta}}^{(\boldsymbol{U})}(\boldsymbol{x}) - f_D^*(\boldsymbol{x}))^2 D_{\boldsymbol{x}}(\boldsymbol{x})d\boldsymbol{x}$.

For the second inequality, since $(a + b)^2 \ge (1 - c)a^2 - (\frac{1}{c} - 1)b^2$ and $\frac{1}{c} - 1 > 0$ for all $c \in (0, 1)$, we have

$$\mathcal{L}_{\text{val}}(f_{\boldsymbol{\theta}}^{(\boldsymbol{U})}) = \frac{1}{2}\mathbb{E}_{(\boldsymbol{x},y)\sim D_{\text{val}}}\left[(f_{\boldsymbol{\theta}}^{(\boldsymbol{U})}(\boldsymbol{x}) - y)^2\right]$$

$$\ge \frac{1-c}{2}\mathbb{E}_{(\boldsymbol{x},y)\sim D_{\text{val}}}\left[(y - f_D^*(\boldsymbol{x}))^2\right] - \frac{\frac{1}{c}-1}{2}\mathbb{E}_{(\boldsymbol{x},y)\sim D_{\text{val}}}\left[(f_{\boldsymbol{\theta}}^{(\boldsymbol{U})}(\boldsymbol{x}) - f_D^*(\boldsymbol{x}))^2\right]$$

$$\ge (1-c)\mathcal{L}_{\text{val}}(f_D^*) - \beta \cdot (\tfrac{1}{c} - 1)\mathcal{L}_D(\boldsymbol{\theta}; \boldsymbol{U}),$$

which completes the proof. $\qquad\square$

Taking the expectation over $\boldsymbol{U}$ and the training process, we have the following corollary.

**Corollary 5.** *For all $m \ge 1$, $\eta > 0$, $B \ge 1$ and $T \ge 1$,*

$$\forall c \in (0, 1]: \quad \left|\mathcal{I}_{\text{val}}^{(m)}(D; \eta, B, T) - \mathcal{L}_{\text{val}}(f_D^*)\right| \le c\mathcal{L}_{\text{val}}(f_D^*) + \beta \cdot (1 + \tfrac{1}{c})\bar{L}_D^{(m)}(\eta, B, T),$$

*where $\bar{L}_D^{(m)}(\eta, B, T) := \mathbb{E}_{(\boldsymbol{\theta},\boldsymbol{U})\sim\mathcal{P}^{(m)}(D;\eta,B,T)}[\mathcal{L}_D(\boldsymbol{\theta}; \boldsymbol{U})]$.*

*Proof.* Taking the expectation over $(\boldsymbol{\theta}, \boldsymbol{U}) \sim \mathcal{P}^{(m)}(D; \eta, B, T)$, we have

$$\mathcal{I}_{\text{val}}^{(m)}(D; \eta, B, T) \le (1 + c)\mathcal{L}_{\text{val}}(f_D^*) + \beta \cdot (1 + \tfrac{1}{c})\bar{L}_D^{(m)}(\eta, B, T), \tag{26}$$

$$\mathcal{I}_{\text{val}}^{(m)}(D; \eta, B, T) \ge (1 - c)\mathcal{L}_{\text{val}}(f_D^*) - \beta \cdot (\tfrac{1}{c} - 1)\bar{L}_D^{(m)}(\eta, B, T). \tag{27}$$

Rearranging the inequalities proves the result. $\qquad\square$

We further decompose $L_D^{(m)}(\eta, B, T)$ as follows:

$$L_D^{(m)}(\eta, B, T) = \underbrace{\mathbb{E}_{\boldsymbol{U}\sim\mathcal{U}^m}[\mathcal{L}_D(\boldsymbol{\mu}; \boldsymbol{U})]}_{\text{approximation error}} + \underbrace{\mathbb{E}_{(\boldsymbol{\theta},\boldsymbol{U})\sim\mathcal{P}^{(m)}(D;\eta,B,T)}[\mathcal{L}_D(\boldsymbol{\theta}; \boldsymbol{U}) - \mathcal{L}_D(\boldsymbol{\mu}; \boldsymbol{U})]}_{\text{optimization error}}. \tag{28}$$

Now we analyze the approximation error and the optimization error separately.

### B.2.2 APPROXIMATION ERROR ANALYSIS

The following lemma gives an upper bound on the approximation error.

**Lemma 6.** *For random feature models with width $m$, it holds that*

$$\mathbb{E}[\mathcal{L}_D(\boldsymbol{\mu}; \boldsymbol{U})] \le \frac{C_1^2}{m}, \tag{29}$$

*where $C_1 := \sup_{\boldsymbol{x}\in\mathcal{X}}\sup_{\boldsymbol{u}\in\text{supp}(\mathcal{U})}\{|\sigma(\langle\boldsymbol{u},\boldsymbol{x}\rangle)| \cdot |\nu(\boldsymbol{u})|\}$.*

*Proof.* Since the distribution $D$ is well-behaved, there exists a function $\nu : \text{supp}(\mathcal{U}) \to \mathbb{R}$ such that $\sup_{\boldsymbol{u}\in\text{supp}(\mathcal{U})}|\nu(\boldsymbol{u})| < \infty$ and it holds for all $\boldsymbol{x} \in \mathcal{X}$ that

$$f_D^*(\boldsymbol{x}) = \mathbb{E}_{\boldsymbol{u}\sim\mathcal{U}}[\sigma(\langle\boldsymbol{u},\boldsymbol{x}\rangle)\nu(\boldsymbol{u})]. \tag{30}$$

Let $\boldsymbol{\theta} \in \mathbb{R}^m$ be the vector whose $i$-th entry is $\frac{1}{\sqrt{m}}\nu(\boldsymbol{u}_i)$. Then, we have

$$\mathbb{E}[\mathcal{L}_D(\boldsymbol{\mu}; \boldsymbol{U})] \leq \mathbb{E}[\mathcal{L}_D(\boldsymbol{\theta}; \boldsymbol{U})] = \mathbb{E}\left[\mathbb{E}_{(\boldsymbol{x},y)\sim\mathcal{D}}\left[\left(y - \frac{1}{m}\sum_{i=1}^{m}\sigma(\langle\boldsymbol{u}_i, \boldsymbol{x}\rangle)\nu(\boldsymbol{u}_i)\right)^2\right]\right] \tag{31}$$

$$= \frac{1}{m}\mathbb{E}_{(\boldsymbol{x},y)\sim\mathcal{D}}\mathrm{Var}_{\boldsymbol{u}\sim\mathcal{U}}\left[\sigma(\langle\boldsymbol{u}, \boldsymbol{x}\rangle)\nu(\boldsymbol{u})\right] \tag{32}$$

$$\leq \frac{C_1^2}{m}, \tag{33}$$

where the last inequality follows from the fact that $\mathrm{Var}_{\boldsymbol{u}\sim\mathcal{U}}\left[\sigma(\langle\boldsymbol{u}, \boldsymbol{x}\rangle)\nu(\boldsymbol{u})\right] \leq C_1^2$. $\qquad\square$

### B.2.3 OPTIMIZATION ERROR ANALYSIS

Now we analyze the optimization error. We first present a series of lemmas that upper bound a few quantities that are related to the optimization error.

**Lemma 7.** *There exists a constant $R > 0$ such that the following conditions hold almost surely over the draw of $\boldsymbol{U}$:*

1. $|\phi^{(\boldsymbol{U})}(\boldsymbol{x})| \leq R$ *for all $\boldsymbol{x} \in \mathcal{X}$;*

2. $\|\phi^{(\boldsymbol{U})}(\boldsymbol{x})\|_2 \leq \sqrt{m}R$ *for all $\boldsymbol{x} \in \mathcal{X}$;*

3. $\boldsymbol{H} \preceq R^2\boldsymbol{I}$;

4. $\mathrm{tr}(\boldsymbol{\Sigma}) \leq R^2\mathcal{L}_D(\boldsymbol{\mu}; \boldsymbol{U})$.

*Proof.* This directly follows from the following facts: (1) each $\boldsymbol{u}_i$ in $\boldsymbol{U}$ is drawn from a bounded distribution $\mathcal{U}$; (2) the input space $\mathcal{X}$ is bounded; (3) the activation function $\sigma$ is Lipschitz. $\qquad\square$

**Lemma 8.** *If $m$ is sufficiently large, there exists a constant $\lambda_0 > 0$ such that $\Pr\left[\lambda_{\min}(\boldsymbol{H}) \geq \lambda_0\right] \geq 1 - C_3 e^{-C_2 m}$ over the draw of $\boldsymbol{U}$.*

*Proof.* Let $\mathcal{T}^{(\boldsymbol{U})} : L^2(D_{\mathrm{x}}) \to L^2(D_{\mathrm{x}})$ be the following integral operator:

$$(\mathcal{T}^{(\boldsymbol{U})}f)(\boldsymbol{x}) := \mathbb{E}_{\boldsymbol{x}'\sim D_{\mathrm{x}}}\left[\frac{1}{m}\sum_{i=1}^{m}\sigma(\langle\boldsymbol{u}_i, \boldsymbol{x}\rangle)\sigma(\langle\boldsymbol{u}_i, \boldsymbol{x}'\rangle)f(\boldsymbol{x}')\right]. \tag{34}$$

Taking the expectation over $\boldsymbol{U}$, we have

$$\mathbb{E}[\mathcal{T}^{(\boldsymbol{U})}f] = \mathcal{T}^{(\infty)}f, \quad \text{where } \mathcal{T}^{(\infty)}f(\boldsymbol{x}) := \mathbb{E}_{\boldsymbol{x}'\sim D_{\mathrm{x}}}\left[K(\boldsymbol{x}, \boldsymbol{x}')f(\boldsymbol{x}')\right]. \tag{35}$$

By applying matrix concentration bounds (Pinelis, 1994) to $\mathcal{T}^{(\boldsymbol{U})} - \mathcal{T}^{(\infty)}$, we have with probability at least $1 - C_3 e^{-C_2 m}$,

$$\|\mathcal{T}^{(\boldsymbol{U})} - \mathcal{T}^{(\infty)}\|_{\mathrm{op}} \leq \frac{1}{2}\lambda_{\min}(\mathcal{T}^{(\infty)}). \tag{36}$$

Combining this with Weyl's inequality and setting $\lambda_0 = \frac{1}{2}\lambda_{\min}(\mathcal{T}^{(\infty)})$ completes the proof. $\qquad\square$

**Lemma 9.** *There exists a constant $\rho > 0$ such that the following conditions hold:*

1. $\mathcal{L}(\boldsymbol{0}; \boldsymbol{U}) \leq \frac{1}{2}\rho^2$;

2. $\|\boldsymbol{\beta}\|_2 \leq R\rho$;

3. $\|\boldsymbol{\mu}\|_2 \leq \frac{R\rho}{\lambda_{\min}(\boldsymbol{H})}$.

*Proof.* Let $\rho^2 := \mathbb{E}_{(\boldsymbol{x},y)\sim\mathcal{D}}\left[y^2\right]$. Then $\mathcal{L}(\boldsymbol{0}; \boldsymbol{U}) = \frac{1}{2}\rho^2$. Since $\|\phi^{(\boldsymbol{U})}(\boldsymbol{x})\|_2 \leq \sqrt{m}R$ for all $\boldsymbol{x} \in \mathcal{X}$, we have $\|\boldsymbol{\beta}\|_2 \leq R\rho$ by Cauchy-Schwarz inequality. By the definition of $\boldsymbol{\mu}$, we have $\|\boldsymbol{\mu}\|_2 \leq \frac{R\rho}{\lambda_{\min}(\boldsymbol{H})}$. $\qquad\square$

Now we are ready to analyze the optimization error. Let $\boldsymbol{\delta}_t := \boldsymbol{\theta}_t - \boldsymbol{\mu}$. The following lemma provides a bound on $\boldsymbol{\delta}_t$.

**Lemma 10.** *Given $\boldsymbol{U} \in \mathbb{R}^{d \times m}$, if $\eta \leq \frac{\lambda_{\min}(\boldsymbol{H})}{R^4}$, then $\mathbb{E}[\|\boldsymbol{\delta}_T\|_2^2 \mid \boldsymbol{U}]$ can be bounded as follows:*

$$\mathbb{E}\left[\|\boldsymbol{\delta}_T\|_2^2 \mid \boldsymbol{U}\right] = (1 - \eta\lambda_{\min}(\boldsymbol{H}))^T \|\boldsymbol{\delta}_0\|_2^2 + \frac{\eta}{B} \cdot \frac{\operatorname{tr}(\Sigma)}{\lambda_{\min}(\boldsymbol{H})}. \tag{37}$$

*Proof.* Let $\boldsymbol{A}_t := \frac{1}{Bm} \sum_{b=1}^B \phi(\boldsymbol{x}_{t,b})\phi(\boldsymbol{x}_{t,b})^\top$ and $\boldsymbol{\xi}_t := \frac{1}{B\sqrt{m}} \sum_{b=1}^B (f_{\boldsymbol{\mu}}^{(\boldsymbol{U})}(\boldsymbol{x}_{t,b}) - y_{t,b})\phi(\boldsymbol{x}_{t,b})$. Then, we can rewrite the update rule as

$$\boldsymbol{\theta}_{t+1} - \boldsymbol{\mu} = \boldsymbol{\theta}_t - \boldsymbol{\mu} - \frac{\eta}{B\sqrt{m}} \sum_{b=1}^B (f_{\boldsymbol{\theta}_t}^{(\boldsymbol{U})}(\boldsymbol{x}_{t,b}) - y_{t,b})\phi(\boldsymbol{x}_{t,b}) \tag{38}$$

$$= \boldsymbol{\theta}_t - \boldsymbol{\mu} - \frac{\eta}{B\sqrt{m}} \sum_{b=1}^B \left( \frac{1}{\sqrt{m}}\langle \boldsymbol{\theta}_t - \boldsymbol{\mu}, \phi(\boldsymbol{x}_{t,b})\rangle + (f_{\boldsymbol{\mu}}^{(\boldsymbol{U})}(\boldsymbol{x}_{t,b}) - y_{t,b}) \right) \phi(\boldsymbol{x}_{t,b}) \tag{39}$$

$$= \boldsymbol{\theta}_t - \boldsymbol{\mu} - \eta\boldsymbol{A}_t(\boldsymbol{\theta}_t - \boldsymbol{\mu}) - \eta\boldsymbol{\xi}_t. \tag{40}$$

Therefore, we have

$$\boldsymbol{\delta}_{t+1} = (\boldsymbol{I} - \eta\boldsymbol{A}_t)\boldsymbol{\delta}_t - \eta\boldsymbol{\xi}_t. \tag{41}$$

Expanding the recursion, we have

$$\boldsymbol{\delta}_t = (\boldsymbol{I} - \eta\boldsymbol{A}_{t-1})\cdots(\boldsymbol{I} - \eta\boldsymbol{A}_0)\boldsymbol{\delta}_0 - \eta\sum_{s=0}^{t-1}(\boldsymbol{I} - \eta\boldsymbol{A}_{t-1})\cdots(\boldsymbol{I} - \eta\boldsymbol{A}_{s+1})\boldsymbol{\xi}_s. \tag{42}$$

Since $\boldsymbol{\xi}_t$ is mean zero and independent of $\boldsymbol{A}_s$ and $\boldsymbol{\xi}_s$ for all $s < t$, we have

$$\mathbb{E}[\|\boldsymbol{\delta}_t\|_2^2 \mid \boldsymbol{U}] = \|(\boldsymbol{I} - \eta\boldsymbol{A}_{t-1})\cdots(\boldsymbol{I} - \eta\boldsymbol{A}_0)\boldsymbol{\delta}_0\|_2^2 \tag{43}$$

$$+ \eta^2 \sum_{s=0}^{t-1} \mathbb{E}\left[\|(\boldsymbol{I} - \eta\boldsymbol{A}_{t-1})\cdots(\boldsymbol{I} - \eta\boldsymbol{A}_{s+1})\boldsymbol{\xi}_s\|_2^2 \mid \boldsymbol{U}\right] \tag{44}$$

Note that for any vector $\boldsymbol{v}$ independent of $\boldsymbol{A}_t$, we have $\mathbb{E}[\|(\boldsymbol{I} - \eta\boldsymbol{A}_t)\boldsymbol{v}\|_2^2] = \mathbb{E}[\boldsymbol{v}^\top(\boldsymbol{I} - 2\eta\boldsymbol{A}_t + \eta^2\boldsymbol{A}_t^2)\boldsymbol{v}] = \|\boldsymbol{v}\|_2^2 - 2\eta\boldsymbol{v}^\top\boldsymbol{H}\boldsymbol{v} + \eta^2 R^4\|\boldsymbol{v}\|_2^2 \leq (1 - 2\eta\lambda_{\min}(\boldsymbol{H}) + \eta^2 R^4)\|\boldsymbol{v}\|_2^2 \leq (1 - \eta\lambda_{\min}(\boldsymbol{H}))\|\boldsymbol{v}\|_2^2$. Then

$$\mathbb{E}[\|\boldsymbol{\delta}_t\|_2^2 \mid \boldsymbol{U}] \leq (1 - \eta\lambda_{\min}(\boldsymbol{H}))^t \|\boldsymbol{\delta}_0\|_2^2 + \eta^2 \sum_{s=0}^{t-1}(1 - \eta\lambda_{\min}(\boldsymbol{H}))^{t-s} \frac{\operatorname{tr}(\Sigma)}{B} \tag{45}$$

$$\leq (1 - \eta\lambda_{\min}(\boldsymbol{H}))^t \|\boldsymbol{\delta}_0\|_2^2 + \frac{\eta}{B} \cdot \frac{\operatorname{tr}(\Sigma)}{\lambda_{\min}(\boldsymbol{H})}, \tag{46}$$

which completes the proof. $\qquad\square$

**Lemma 11.** *There exists a constant $C_4, C_5, C_6 > 0$ such that for all $\eta \leq \frac{\lambda_0}{R^4}$,*

$$\mathbb{E}[\mathcal{L}_D(\boldsymbol{\theta}; \boldsymbol{U}) - \mathcal{L}_D(\boldsymbol{\mu}; \boldsymbol{U})] \leq C_4(1 - \eta\lambda_0)^T + \frac{C_5\eta}{B}\mathbb{E}[\mathcal{L}_D(\boldsymbol{\mu}; \boldsymbol{U})] + C_6 e^{-C_2 m}, \tag{47}$$

*where the expectation is taken over $(\boldsymbol{\theta}, \boldsymbol{U}) \sim \mathcal{P}^{(m)}(D; \eta, B, T)$.*

*Proof.* With probability at least $1 - C_3 e^{-C_2 m}$, we have $\lambda_{\min}(\boldsymbol{H}) \geq \lambda_0$. By Lemma 10, we have

$$\mathbb{E}\left[\|\boldsymbol{\delta}_T\|_2^2 \mid \boldsymbol{U}\right] = (1 - \eta\lambda_0)^T \|\boldsymbol{\delta}_0\|_2^2 + \frac{\eta}{B} \cdot \frac{\operatorname{tr}(\Sigma)}{\lambda_0} \tag{48}$$

$$\leq (1 - \eta\lambda_0)^T \left(\frac{R\rho}{\lambda_0}\right)^2 + \frac{\eta}{B} \cdot \frac{1}{\lambda_0} \cdot R^2 \mathcal{L}_D(\boldsymbol{\mu}; \boldsymbol{U}). \tag{49}$$

Let $C_4 := R^2 \cdot \left(\frac{R\rho}{\lambda_0}\right)^2$ and $C_5 := R^2 \cdot \frac{R^2}{\lambda_0}$. Then we have

$$\mathbb{E}[\mathcal{L}_D(\boldsymbol{\theta};\boldsymbol{U}) - \mathcal{L}_D(\boldsymbol{\mu};\boldsymbol{U}) \mid \boldsymbol{U}] \leq R^2 \mathbb{E}\left[\|\boldsymbol{\delta}_T\|_2^2 \mid \boldsymbol{U}\right] \tag{50}$$

$$\leq C_4 \left(1 - \eta\lambda_0\right)^T + \frac{C_5\eta}{B}\mathcal{L}_D(\boldsymbol{\mu};\boldsymbol{U}). \tag{51}$$

If $\lambda_{\min}(\boldsymbol{H}) < \lambda_0$, then we still have $\mathcal{L}_D(\boldsymbol{\mu};\boldsymbol{U}) \leq C_6$ for some constant $C_6' > 0$ since $\eta \leq \frac{1}{\lambda_{\max}(\boldsymbol{H})}$ and descent lemma holds. Let $C_6 := C_6'C_3$. Putting all the pieces together proves the result. $\qquad\square$

### B.2.4 PUTTING LEMMAS TOGETHER

*Proof of Lemma 3.* By Corollary 5, it holds for all $c \in (0, 1]$ that

$$\left|\mathcal{I}_{\mathrm{val}}^{(m)}(D;\eta,B,T) - \mathcal{L}_{\mathrm{val}}(f_D^*)\right| \leq c\mathcal{L}_{\mathrm{val}}(f_D^*) + \beta \cdot (1 + \tfrac{1}{c})\bar{L}_D^{(m)}(\eta,B,T). \tag{52}$$

By Lemmas 6 and 11, we can bound $\bar{L}_D^{(m)}(\eta,B,T)$ by

$$\bar{L}_D^{(m)}(\eta,B,T) \leq \mathbb{E}[\mathcal{L}_D(\boldsymbol{\mu};\boldsymbol{U})] + C_4(1 - \eta\lambda_0)^T + \frac{C_5\eta}{B}\mathbb{E}[\mathcal{L}_D(\boldsymbol{\mu};\boldsymbol{U})] + C_6e^{-C_2m} \tag{53}$$

$$\leq \frac{C_1^2C_5\eta}{Bm} + C_4(1 - \eta\lambda_0)^T + \frac{C_1^2}{m} + C_6e^{-C_2m}. \tag{54}$$

Let $\delta$ be a parameter. Let $\eta_{\max} := \frac{\lambda_0}{R^4}$. If $m$ is so large that $\frac{C_1^2}{m} + C_6e^{-C_2m} \leq \frac{\delta}{4}$, and if $T$ is so large that $C_4(1 - \eta_{\max}\lambda_0)^T \leq \frac{\delta}{4}$, then if $\eta$ is so small that $\frac{C_1^2C_5\eta}{Bm} \leq \frac{\delta}{4}$ and is not too small to make $C_4(1 - \eta\lambda_0)^T$ larger than $\frac{\delta}{4}$, we have

$$\bar{L}_D^{(m)}(\eta,B,T) \leq \delta. \tag{55}$$

Setting a small enough $\delta$, we have

$$\left|\mathcal{I}_{\mathrm{val}}^{(m)}(D;\eta,B,T) - \mathcal{L}_{\mathrm{val}}(f_D^*)\right| \leq c\mathcal{L}_{\mathrm{val}}(f_D^*) + \beta \cdot (1 + \tfrac{1}{c})\delta \tag{56}$$

$$\leq \beta\delta + 2\sqrt{\beta\mathcal{L}_{\mathrm{val}}(f_D^*)\delta}, \tag{57}$$

where we set $c = \sqrt{\frac{\beta\delta}{\mathcal{L}_{\mathrm{val}}(f_D^*)}}$. Finally, we can make $\delta$ so small that the right hand side is smaller than $\Delta$, which completes the proof. $\qquad\square$

## C  AN EMPIRICAL CHARACTERIZATION OF "TINY" LEARNING RATES (EXTENDED)

Our analysis suggests the potential benefits of training proxy models with tiny learning rates. A natural question is *how small should the learning rate be to ensure reliable transferability?* We provide both theoretical guidance and practical recommendations here.

**Theoretical guidance.** While our main analysis focuses on the full-batch gradient descent case, real-world training uses SGD. Here, we extend our analysis to establish precise bounds on what constitutes a "tiny" learning rate when mini-batch SGD introduces gradient noise. We stress that this one-step analysis is standard in the literature (e.g. McCandlish et al. (2018); Smith et al. (2018)) and has often been shown to give quantitatively useful guidance when coupled with empirical validation. For a single SGD update, the parameter change is:

$$\theta_1 = \theta - \eta(\nabla\ell(\theta) + \xi)$$

where $\xi$ is zero-mean gradient noise with covariance $\Sigma/B$, $B$ is the batch size. We analyze the expected validation loss change through a second-order Taylor expansion:

$$\Delta\ell_{\text{val}}(\theta) \approx -\eta\nabla\ell_{\text{val}}(\theta) \cdot (\nabla\ell(\theta) + \xi) + \frac{\eta^2}{2}(\nabla\ell(\theta) + \xi)^T H(\nabla\ell(\theta) + \xi)$$

where $H := H_{\ell_{\text{val}}}$ is the validation loss Hessian. We introduce shorthand notation $a = \nabla\ell_{\text{val}}(\theta) \cdot \nabla\ell(\theta)$ for the gradient alignment and $b = \nabla\ell(\theta)^T H\nabla\ell(\theta)$ for the curvature term. Expanding the second-order Taylor approximation, we have

$$\Delta\ell_{\text{val}}(\theta) \approx -\eta a - \eta\nabla\ell_{\text{val}}(\theta) \cdot \xi + \frac{\eta^2}{2}\left(b + 2\xi^T H\nabla\ell(\theta) + \xi^T H\xi\right)$$

Taking expectation over mini-batch noise $\xi$:

$$\mathbb{E}[\Delta\ell_{\text{val}}] = -\eta a + \frac{\eta^2}{2}\left(b + \mathbb{E}[\xi^T H\xi]\right)$$

For the quadratic term, we have $\mathbb{E}[\xi^T H\xi] = \mathbb{E}[\text{tr}(H\xi\xi^T)] = \text{tr}(H\mathbb{E}[\xi\xi^T]) = \text{tr}(H\Sigma)/B$, yielding:

$$\mathbb{E}[\Delta\ell_{\text{val}}] = -\eta a + \frac{\eta^2}{2}\left(b + \frac{\text{tr}(H\Sigma)}{B}\right)$$

For dataset quality assessment to remain consistent, the first-order term must dominate the gradient dynamics. Consider two datasets $D_i$ and $D_j$ with corresponding values $a_i, b_i, \Sigma_i$ and $a_j, b_j, \Sigma_j$. The datasets will maintain their relative ordering based on the first-order terms when:

$$|\eta(a_i - a_j)| \gg \frac{\eta^2}{2}\left|b_i - b_j + \frac{1}{B}[\text{tr}(H\Sigma_i) - \text{tr}(H\Sigma_j)]\right|$$

Rearranging to solve for $\eta$:

$$\eta \ll \frac{2|a_i - a_j|}{\left|b_i - b_j + \frac{1}{B}[\text{tr}(H\Sigma_i) - \text{tr}(H\Sigma_j)]\right|}$$

To find a uniform bound valid for all dataset pairs, we define:

$$\Delta a := \min_{i \neq j}|a_i - a_j| \quad \text{(smallest alignment gap between two datasets)}$$

$$G^2 := \max_k \|\nabla\ell_k(\theta)\|^2 \quad \text{(maximum squared gradient norm)}$$

$$\sigma_g^2 := \max_k \text{tr}(\Sigma_k) \quad \text{(maximum gradient trace variance)}$$

For the curvature term difference, we have:

$$|b_i - b_j| = |\nabla\ell_i(\theta)^T H\nabla\ell_i(\theta) - \nabla\ell_j(\theta)^T H\nabla\ell_j(\theta)|$$
$$\leq \lambda_{\max}(\|\nabla\ell_i(\theta)\|^2 + \|\nabla\ell_j(\theta)\|^2)$$
$$\leq 2\lambda_{\max}G^2$$

where $\lambda_{\max} = \|H\|_2$ is the spectral norm of the Hessian. Similarly, for the noise term difference:

$$\left| \frac{1}{B}[\mathrm{tr}(H\Sigma_i) - \mathrm{tr}(H\Sigma_j)] \right| \leq \frac{\lambda_{\max}}{B}[\mathrm{tr}(\Sigma_i) + \mathrm{tr}(\Sigma_j)] \leq \frac{2\lambda_{\max}\sigma_g^2}{B}$$

Combining these bounds and applying the triangle inequality:

$$\left| b_i - b_j + \frac{1}{B}[\mathrm{tr}(H\Sigma_i) - \mathrm{tr}(H\Sigma_j)] \right| \leq 2\lambda_{\max}\left(G^2 + \frac{\sigma_g^2}{B}\right)$$

This yields our final upper bound on the learning rate:

$$\eta_{\mathrm{tiny}} \leq \frac{\Delta a}{\lambda_{\max}\left(G^2 + \frac{\sigma_g^2}{B}\right)} \tag{58}$$

**Remark: Scope of the analysis.** Our bound in Equation (58) provides local guidance by controlling the expected loss change for the next SGD step. However, practical training involves multiple steps where parameters evolve continuously, and higher-order terms along with ill-conditioning effects can become significant. Following established practices in analyzing *critical batch size* (CBS) (McCandlish et al., 2018), we treat this bound as theoretical guidance rather than a strict constraint. **Empirical validation.** Figure 7 and 8 demonstrate that our theoretically predicted values of $\eta_{\mathrm{tiny}}$ align well with empirically observed regimes of high cross-scale transferability. We estimate $\eta_{\mathrm{tiny}}$ using model checkpoints obtained after short warmup training. Specifically, we train each candidate dataset for 500 warmup steps using a standard learning rate of $3 \times 10^{-4}$ Karpathy (2022), with complete training configurations detailed in Appendix D.1. We then compute the upper bound from Equation (58) using statistics gathered from 20 batches sampled during this warmup phase. Our bound estimation requires computing Hessian eigenvalues and gradient variances, which we estimate efficiently through power iteration combined with Pearlmutter's Hessian-vector products and a covariance trace Monte Carlo estimator. Importantly, this estimation process requires only minutes of computation time, making it practically viable for real-world applications. As shown in both Figure 7 and 8, the predicted values consistently fall within the learning rate regime that exhibits very high cross-scale transferability, thereby validating both the theoretical soundness and practical utility of our analysis.

**Choosing $\eta_{\mathrm{tiny}}$ in practice: a simple rule of thumb.** While estimating $\eta_{\mathrm{tiny}}$ from proxy model checkpoints is computationally inexpensive, we find that a simpler heuristic suffices for most LLM pretraining scenarios. In typical LLM pretraining, gradient clipping is used to control gradient magnitudes, with the batch gradient norm capped at some constant $C$. We can therefore conservatively set $G = C$. Substituting these values into our bound yields $\eta_{\mathrm{tiny}} < \frac{\Delta a}{\lambda_{\max}C^2}$. Since optimal learning rates for neural network training typically scale as $1/\lambda_{\max}$ (LeCun et al., 2002), $\eta_{\mathrm{tiny}}$ should be orders of magnitude smaller than the optimal learning rate, with the specific factor depending on the minimum gradient alignment difference between any dataset pair. Empirically, we find that *using learning rates 1-2 orders of magnitude smaller than optimal* generally works well. For language model pretraining, this typically translates to learning rates of approximately $10^{-5}$ to $10^{-6}$ Radford et al. (2019); Karpathy (2022). As demonstrated in Figure 7 and 8, these predicted values consistently achieve high cross-scale transferability across various model architectures and scales. Importantly, while such small learning rates would be impractical for complete model training due to slow convergence, they prove highly effective in assessing and comparing different data mixtures.

**Lower limit of $\eta_{\mathrm{tiny}}$.** We further remark that in practice, $\eta_{\mathrm{tiny}}$ can also not be too small due to factors such as floating point errors. While extremely small learning rates benefit transferability, there exists a practical lower bound determined by finite floating point precision. In 32-bit floating-point format, every number is represented with a 23-bit mantissa, giving around 7 decimal digits of precision. The gap between 1.0 and the next larger representable value is therefore $2^{-23} \approx 1.19 \cdot 10^{-7}$, where any increment smaller than this is rounded away. Consequently, for parameters whose magnitude is $O(1)$, the parameter update magnitude (i.e., learning-rate-times-gradient product) below $10^{-7}$ will be flushed to zero, imposing a practical lower bound on usable $\eta_{\mathrm{tiny}}$.

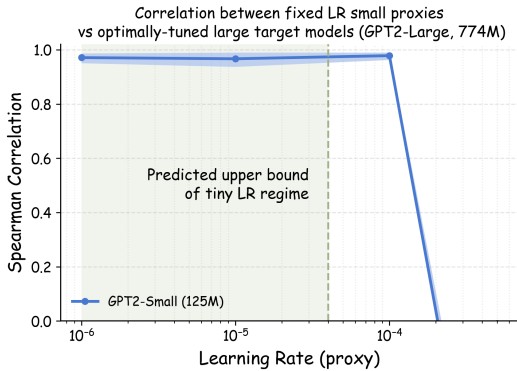

Figure 7: The same GPT2-Small curve in Figure 6 (a), but highlighting the theoretical bounds of tiny learning rate regimes. The shaded region indicates the theoretically predicted range for $\eta_{\text{tiny}}$, with the upper bound given by Equation (58).

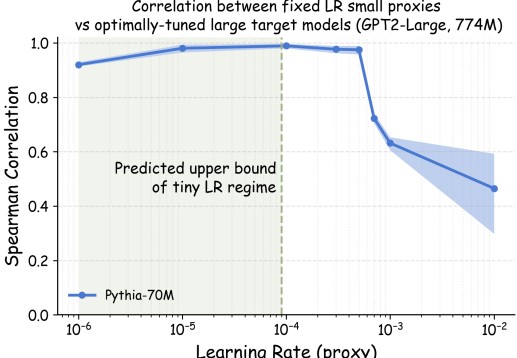

Figure 8: The same Pythia-70M curve in Figure 6 (a) but highlight the theoretical bounds of tiny learning rate regimes. The shaded region indicates the theoretically predicted range for $\eta_{\text{tiny}}$, with the upper bound given by Equation (58).

# D  EXPERIMENT SETTINGS & ADDITIONAL EXPERIMENTS

## D.1  ADDITIONAL EXPERIMENT SETTINGS

### D.1.1  DETAILS FOR EVALUATION METRICS

**Top-$k$ Decision Regret.** To quantify the practical risk of proxy-model-based dataset selection, we define the top-$k$ decision regret as the performance gap between the truly optimal dataset and the best-performing dataset among the proxy's top-$k$ recommendations. Let $\mathcal{S}_k$ denote the set of $k$ datasets with the highest small proxy model performance. The top-$k$ decision regret is then expressed as:

$$\Delta_{\text{opt}}^{(k)} = \min_{D_i \in \mathcal{S}_k, \lambda \in \Lambda} \ell_{\text{val}}(\theta_{\text{tgt}}(D_i; \lambda)) - \min_{D_j \in \mathcal{D}, \lambda \in \Lambda} \ell_{\text{val}}(\theta_{\text{tgt}}(D_j; \lambda))$$

where the first term represents the best achievable performance (in terms of validation loss) among the proxy's top-k selections when each is tuned on the large target model, and the second term represents the globally optimal performance across all candidate datasets. This metric captures the performance degradation that could result from relying on proxy model rankings, providing an estimate of the practical risk in dataset selection workflows.

### D.1.2  MODEL TRAINING AND HYPERPARAMETER TUNING

**Model architectures.** We evaluate on 3 classes of different model architectures: GPT-2 (Radford et al., 2019), Pythia (Biderman et al., 2023), and OPT (Zhang et al., 2022). For target models, we use GPT2-Large (774M) and Pythia-1B. For small proxy models, we use GPT2-Small (125M), Pythia-70M, and OPT-125M.

**Training configurations and hyperparameter tuning.** For all training runs, we set token-per-parameter (TPP) ratio to be 20, i.e., $1\times$ Chinchilla's compute-optimal ratio (Hoffmann et al., 2022) with the exception of the ablation study in Appendix D.2.2. We use AdamW optimizer (Loshchilov & Hutter, 2017) across all training runs. For hyperparameter tuning on large target models, we use simple grid search across the three most important hyperparameters in LLM pretraining: learning rate, batch size, and weight decay. For learning rate, we sweep from $8 \times 10^{-5}$, $1 \times 10^{-4}$, $3 \times 10^{-4}$, $5 \times 10^{-4}$, $7 \times 10^{-4}$, $1 \times 10^{-3}$. For batch size, we sweep from 128, 256, 512, 1024. For weight decay, we sweep from 0.001, 0.01, 0.1. For small proxy models, we set batch size as 128 and weight decay as 0.1, with additional ablation studies on other choices of these two hyperparameters in Appendix D.2.2. For both small proxy and large target models, we set other hyperparameters in the AdamW optimizer to be $\beta_1 = 0.9$, $\beta_2 = 0.999$, and $\epsilon = 1e - 8$. We fix the context length at 1024 tokens across all training runs. We follow the Warmup-Stable-Decay (WSD) learning rate scheduler proposed in MiniCPM Hu et al. (2024), allocating 2000 training steps for warmup, 66% for stable training at peak learning rate, and the remaining for linear decay. Throughout this work, "learning rate" refers to the peak learning rate achieved during the stable phase of the WSD schedule. We use full precision training for small proxy models and bfloat16 precision for large target models. We set gradient clipping with a maximum norm of 1.0 to stabilize training dynamics across all training runs. All model training is conducted on 32 NVIDIA H100 GPUs with 80 GB of memory.

### D.1.3  DOWNSTREAM BENCHMARK EVALUATION

We use the Language Model Evaluation Harness Gao et al. (2024) for benchmark evaluation. For each benchmark, the framework computes the length-normalized log-likelihood of the model on each candidate answer and marks a response as correct if the correct answer receives the highest log-likelihood score.

### D.1.4  DETAILED DATASET RECIPE CONFIGURATIONS

We evaluate our tiny learning rate approach across 23 carefully designed data recipes that span a wide spectrum of practical data curation decisions faced by LLM pretraining practitioners. The detailed description for the data recipes considered in this work is summarized in Table 1.

**Category 1: Domain composition and ablation.** We create 10 domain mixture variations from The Pile (Gao et al., 2020). Specifically, 4 recipes maintain a fixed 70% Pile-CC allocation while

Table 1: Summary of 23 dataset recipes used in our experiments. Recipes are grouped by category to represent different data curation decisions faced in practice.

| Category | Recipe ID | Description |
|---|---|---|
| Domain Composition | `Pile-CC70-S0A30` | 70% Pile-CC, 0% StackExchange, 30% ArXiv |
| | `Pile-CC70-S10A20` | 70% Pile-CC, 10% StackExchange, 20% ArXiv |
| | `Pile-CC70-S20A10` | 70% Pile-CC, 20% StackExchange, 10% ArXiv |
| | `Pile-CC70-S30A0` | 70% Pile-CC, 30% StackExchange, 0% ArXiv |
| | `Pile-noArXiv` | Full Pile excluding ArXiv |
| | `Pile-noGitHub` | Full Pile excluding GitHub |
| | `Pile-noEuroParl` | Full Pile excluding EuroParl |
| | `Pile-noStackExchange` | Full Pile excluding StackExchange |
| | `Pile-noPubMedCentral` | Full Pile excluding PubMed Central |
| | `Pile-noWikipedia` | Full Pile excluding Wikipedia |
| Established Datasets | `C4` | C4 (Raffel et al., 2020) |
| | `DCLM` | DCLM-baseline (Li et al., 2024) |
| | `RefinedWeb` | RefinedWeb (Penedo et al., 2023) |
| Quality Filter | `RP2-H50T50` | 50% head-middle, 50% tail |
| | `RP2-H60T40` | 60% head-middle, 40% tail |
| | `RP2-H70T30` | 70% head-middle, 30% tail |
| | `RP2-H80T20` | 80% head-middle, 20% tail |
| | `RP2-H90T10` | 90% head-middle, 10% tail |
| | `RP2-H100T0` | 100% head-middle, 0% tail |
| Deduplication strength | `DCLM-dedup-L` | Strict line/paragraph thresholds |
| | `DCLM-dedup-LS` | Very strict line/paragraph thresholds |
| | `DCLM-dedup-G` | Strict 2,3,4-gram thresholds |
| | `DCLM-dedup-GS` | Strict 5,6,7,8,9,10-gram thresholds |

varying the remaining 30% between StackExchange and ArXiv. Additionally, 6 recipes are created by excluding one domain from the full Pile dataset to assess domain importance.

**Category 2: Established datasets.** We evaluate 3 widely-adopted pretraining corpora: C4 (Raffel et al., 2020), DCLM-baseline (Li et al., 2024), and RefinedWeb (Penedo et al., 2023). They all originate from the Common Crawl but vary in their curation strategies, including data filtering criteria and deduplication algorithms.

**Category 3: Scoring-based data filter.** We construct 6 data recipes using different mixing ratios of head-middle versus tail partitions from RedPajama-V2 Weber et al. (2024), ranging from 50:50 to 100:0. These partitions are defined by the perplexity scores from a 5-gram Kneser-Ney model trained on Wikipedia (Wenzek et al., 2020). Specifically, RedPajama-V2 (Weber et al., 2024) partitions its 113B documents into quality buckets based on the perplexity scores from a 5-gram Kneser-Ney model trained on Wikipedia corpus. Documents are categorized as "head" (low perplexity), "middle" (medium perplexity), or "tail" (high perplexity). Our six mixing ratios between the head-middle and tail partitions represent different decision-making in score-based data filtering faced by LLM pretraining practitioners.

**Category 4: Deduplication.** To understand how deduplication decisions transfer across scales, we create 4 variants of DCLM-baseline by modifying the Massive Web repetition filter Rae et al. (2021) used in its curation pipeline. The Massive Web repetition filter removes documents with excessive n-gram repetitions at various granularities. We systematically adjust thresholds for line/paragraph repetitions and n-gram frequencies to create 4 new data recipes. The default thresholds for the Massive Web repetition filter are summarized in Table 2. We further summarize the changes in thresholds for each of the newly created 4 data recipes in Tables 3, 4, 5, and 6. At a high level, the default thresholds are relatively conservative, and the newly created 4 data recipes are more aggressive in removing documents with repeated lines (`DCLM-dedup-L`and `DCLM-dedup-LS`) or short n-grams (`DCLM-dedup-G`and `DCLM-dedup-GS`).

Table 2: Default Massive Web repetition thresholds (used in `DCLM`). A document is filtered out if any measurement exceeds its threshold; lower is stricter.

| Measurement | Threshold |
|---|---|
| Duplicate line fraction | 0.30 |
| Duplicate paragraph fraction | 0.30 |
| Duplicate line character fraction | 0.20 |
| Duplicate paragraph character fraction | 0.20 |
| Top 2-gram character fraction | 0.20 |
| Top 3-gram character fraction | 0.18 |
| Top 4-gram character fraction | 0.16 |
| Duplicate 5-gram character fraction | 0.15 |
| Duplicate 6-gram character fraction | 0.14 |
| Duplicate 7-gram character fraction | 0.13 |
| Duplicate 8-gram character fraction | 0.12 |
| Duplicate 9-gram character fraction | 0.11 |
| Duplicate 10-gram character fraction | 0.10 |

Table 3: Massive Web repetition thresholds for `DCLM-dedup-L`. Changes vs. baseline in **bold**.

| Measurement | Threshold |
|---|---|
| Duplicate line fraction | **0.20** |
| Duplicate paragraph fraction | **0.20** |
| Duplicate line character fraction | **0.15** |
| Duplicate paragraph character fraction | **0.15** |
| Top 2-gram character fraction | 0.20 |
| Top 3-gram character fraction | 0.18 |
| Top 4-gram character fraction | 0.16 |
| Duplicate 5-gram character fraction | 0.15 |
| Duplicate 6-gram character fraction | 0.14 |
| Duplicate 7-gram character fraction | 0.13 |
| Duplicate 8-gram character fraction | 0.12 |
| Duplicate 9-gram character fraction | 0.11 |
| Duplicate 10-gram character fraction | 0.10 |

Table 4: Massive Web repetition thresholds for `DCLM-dedup-LS`. Changes vs. baseline in **bold**.

| Measurement | Threshold |
|---|---|
| Duplicate line fraction | **0.15** |
| Duplicate paragraph fraction | **0.15** |
| Duplicate line character fraction | **0.10** |
| Duplicate paragraph character fraction | **0.10** |
| Top 2-gram character fraction | 0.20 |
| Top 3-gram character fraction | 0.18 |
| Top 4-gram character fraction | 0.16 |
| Duplicate 5-gram character fraction | 0.15 |
| Duplicate 6-gram character fraction | 0.14 |
| Duplicate 7-gram character fraction | 0.13 |
| Duplicate 8-gram character fraction | 0.12 |
| Duplicate 9-gram character fraction | 0.11 |
| Duplicate 10-gram character fraction | 0.10 |

Table 5: Massive Web repetition thresholds for `DCLM-dedup-G` (tightening 2–4-gram tests). Changes vs. baseline in **bold**.

| Measurement | Threshold |
|---|---|
| Duplicate line fraction | 0.30 |
| Duplicate paragraph fraction | 0.30 |
| Duplicate line character fraction | 0.20 |
| Duplicate paragraph character fraction | 0.20 |
| Top 2-gram character fraction | **0.14** |
| Top 3-gram character fraction | **0.12** |
| Top 4-gram character fraction | **0.10** |
| Duplicate 5-gram character fraction | 0.15 |
| Duplicate 6-gram character fraction | 0.14 |
| Duplicate 7-gram character fraction | 0.13 |
| Duplicate 8-gram character fraction | 0.12 |
| Duplicate 9-gram character fraction | 0.11 |
| Duplicate 10-gram character fraction | 0.10 |

Table 6: Massive Web repetition thresholds for `DCLM-dedup-GS` (tightening 5–10-gram tests). Changes vs. baseline in **bold**.

| Measurement | Threshold |
|---|---|
| Duplicate line fraction | 0.30 |
| Duplicate paragraph fraction | 0.30 |
| Duplicate line character fraction | 0.20 |
| Duplicate paragraph character fraction | 0.20 |
| Top 2-gram character fraction | 0.20 |
| Top 3-gram character fraction | 0.18 |
| Top 4-gram character fraction | 0.16 |
| Duplicate 5-gram character fraction | **0.08** |
| Duplicate 6-gram character fraction | **0.07** |
| Duplicate 7-gram character fraction | **0.06** |
| Duplicate 8-gram character fraction | **0.05** |
| Duplicate 9-gram character fraction | **0.04** |
| Duplicate 10-gram character fraction | **0.03** |

### D.2 ADDITIONAL RESULTS & DISCUSSION

#### D.2.1 ADDITIONAL RESULTS FOR SECTION 6.2

This section presents supplementary results that extend the findings reported in Section 6.2. Figure 9 to Figure 15 demonstrate the top-k decision regret performance when using alternative proxy architectures (Pythia-70M and OPT-125M) when GPT2-Large is the target model. These results corroborate our main findings by showing that the tiny learning rate approach consistently outperforms standard learning rates across diverse model families. In Figure 11, 12 and 13 we present the top-k decision regret decomposed across specific evaluation tasks, including HellaSwag, Winogrande, OpenBookQA, ARC-Easy, and CommonsenseQA, as well as the aggregate performance across all benchmarks. Figure 16 to 22 replicates the complete experimental protocol using Pythia-1B as the target architecture. The consistency of improvements across different evaluation metrics and proxy/target model architectures strengthens the practical applicability of our method in real-world deployment scenarios.

**Proxy training runs with tiny learning rates exhibit significantly smaller stochasticity.** In addition to achieving superior rank correlation, proxy models trained with tiny learning rates demonstrate greatly reduced variance in transferability. As we can see in Figures 6 (a) and 16, the variation in rank correlation due to training randomness is substantially narrower for learning rates below $1 \times 10^{-4}$. This phenomenon can be attributed to the fact that tiny learning rates produce smaller parameter updates, making the training process less sensitive to stochastic factors. More specifically, a very small learning rate produces proportionally smaller parameter diffusions, reducing sensitivity to stochastic factors such as random initialization and gradient noise (jastrzkebski et al., 2017; Stephan et al., 2017). This stability advantage of tiny learning rates provides an additional practical benefit: practitioners can achieve consistent dataset rankings without requiring multiple training runs to account for stochastic variation.

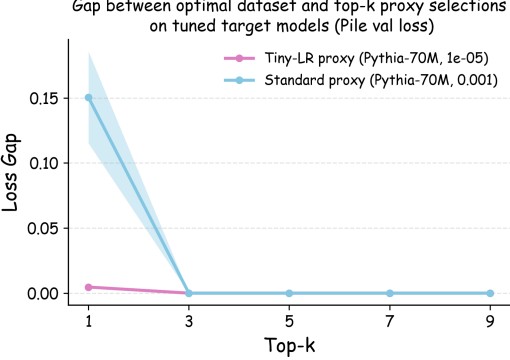

Figure 9: Top-$k$ decision regret between the optimal dataset and the best dataset among top-$k$ proxy (Pythia-70M) selections on hyperparameter-optimized target models (GPT2-Large), measured using Pile validation loss. Shaded areas show 95% bootstrap CIs computed by resampling seeds (3 runs per dataset).

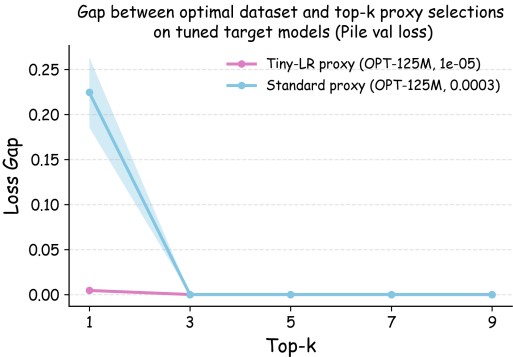

Figure 10: Top-$k$ decision regret between the optimal dataset and the best dataset among top-$k$ proxy (OPT-125M) selections on hyperparameter-optimized target models (GPT2-Large), measured using Pile validation loss. Shaded areas show 95% bootstrap CIs computed by resampling seeds (3 runs per dataset).

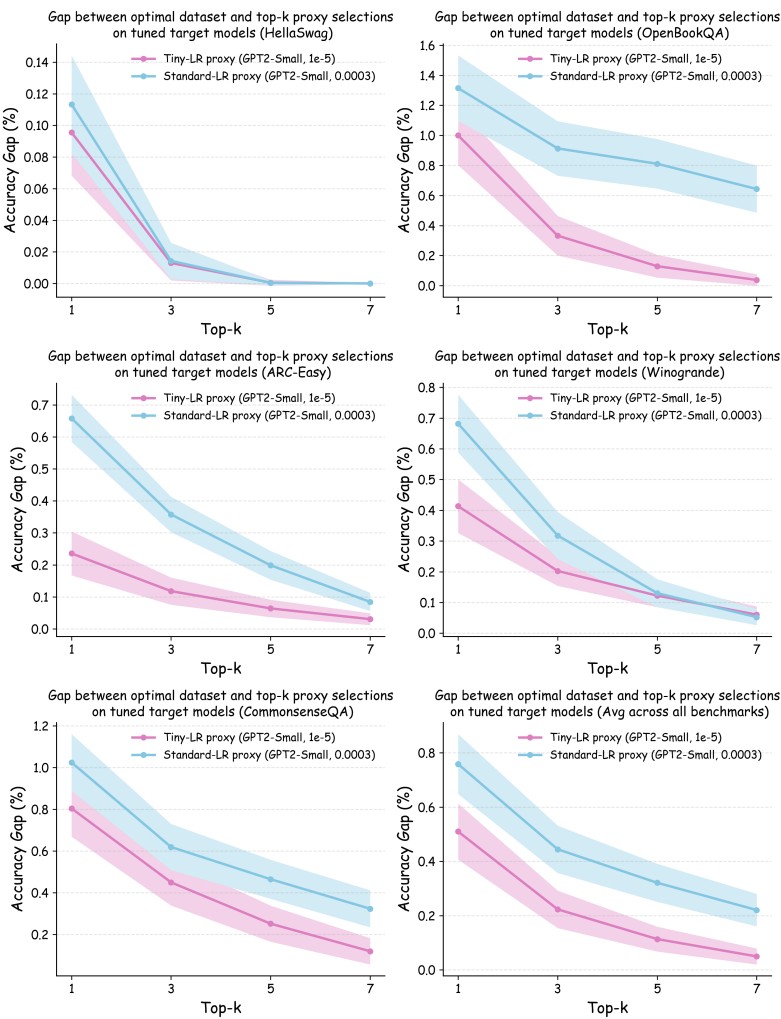

Figure 11: Top-$k$ decision regret between the optimal dataset and the best dataset among top-$k$ proxy (GPT2-Small) selections on hyperparameter-optimized target models (GPT2-Large), measured using downstream benchmarks.

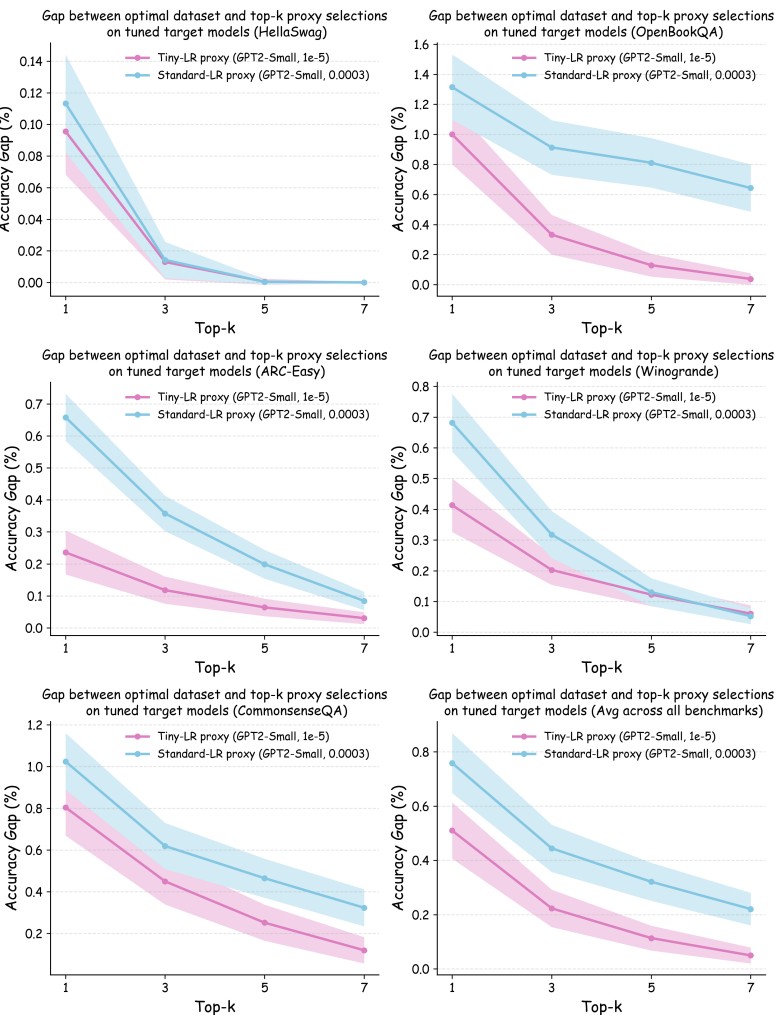

Figure 12: Top-$k$ decision regret between the optimal dataset and the best dataset among top-$k$ proxy (Pythia-70M) selections on hyperparameter-optimized target models (GPT2-Large), measured using downstream benchmarks.

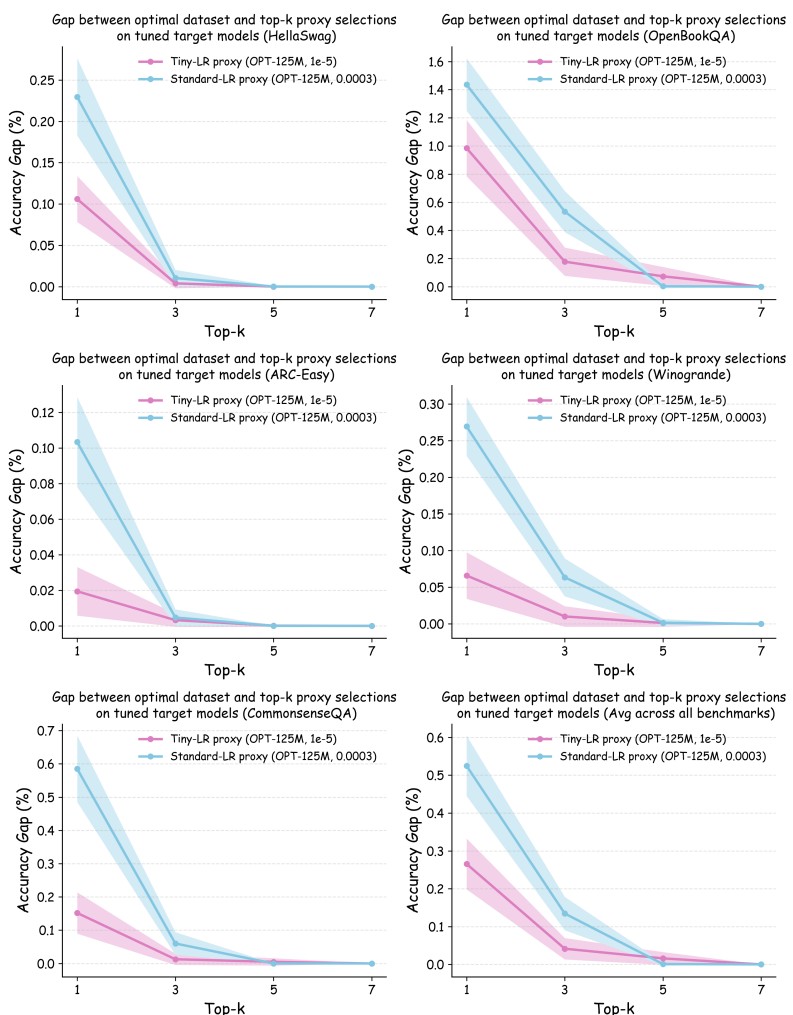

Figure 13: Top-$k$ decision regret between the optimal dataset and the best dataset among top-$k$ proxy (OPT-125M) selections on hyperparameter-optimized target models (GPT2-Large), measured using downstream benchmarks.

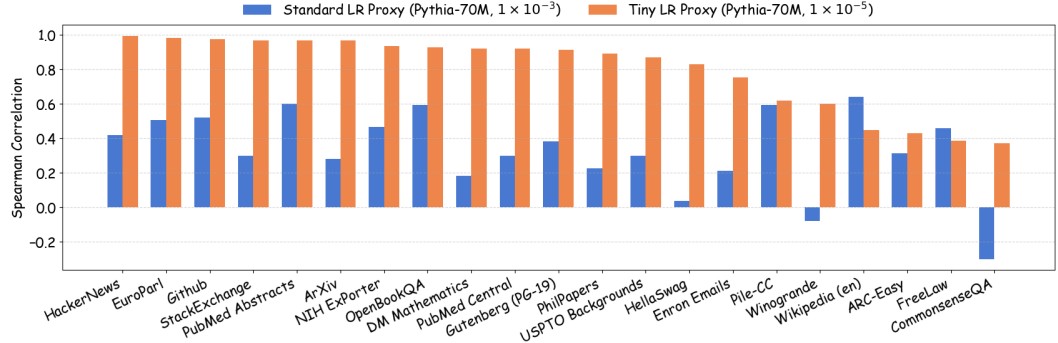

Figure 14: Average rank correlation between proxy (Pythia-70M) and target (GPT2-Large) for the loss computed over a variety of validation domains (from Pile) and downstream benchmarks.

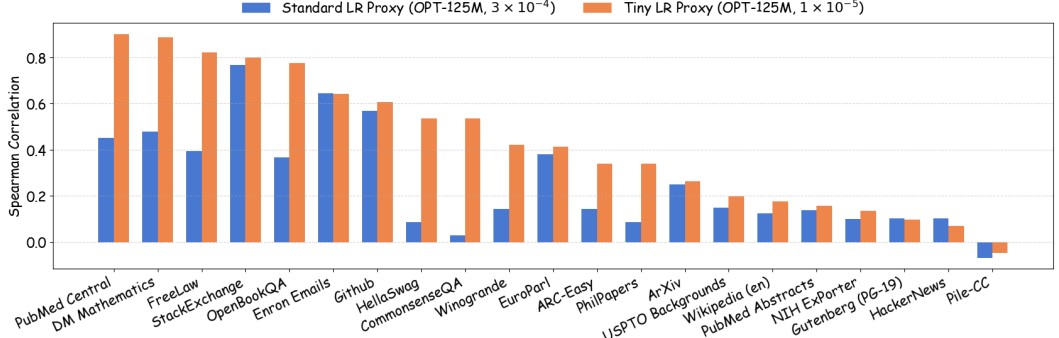

Figure 15: Average rank correlation between proxy (OPT-125M) and target (GPT2-Large) for the loss computed over a variety of validation domains (from Pile) and downstream benchmarks.

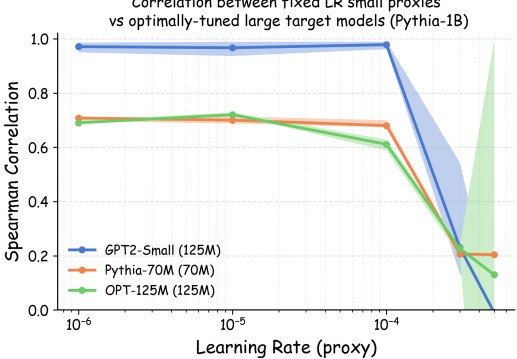

Figure 16: Spearman rank correlation between dataset rankings from proxy models (GPT2-Small, Pythia-70M, OPT-125M) and larger target model (Pythia-1B) as a function of proxy model learning rate, evaluated on aggregated Pile validation loss. Shaded areas show 95% bootstrap CIs computed by resampling seeds (3 runs per dataset).

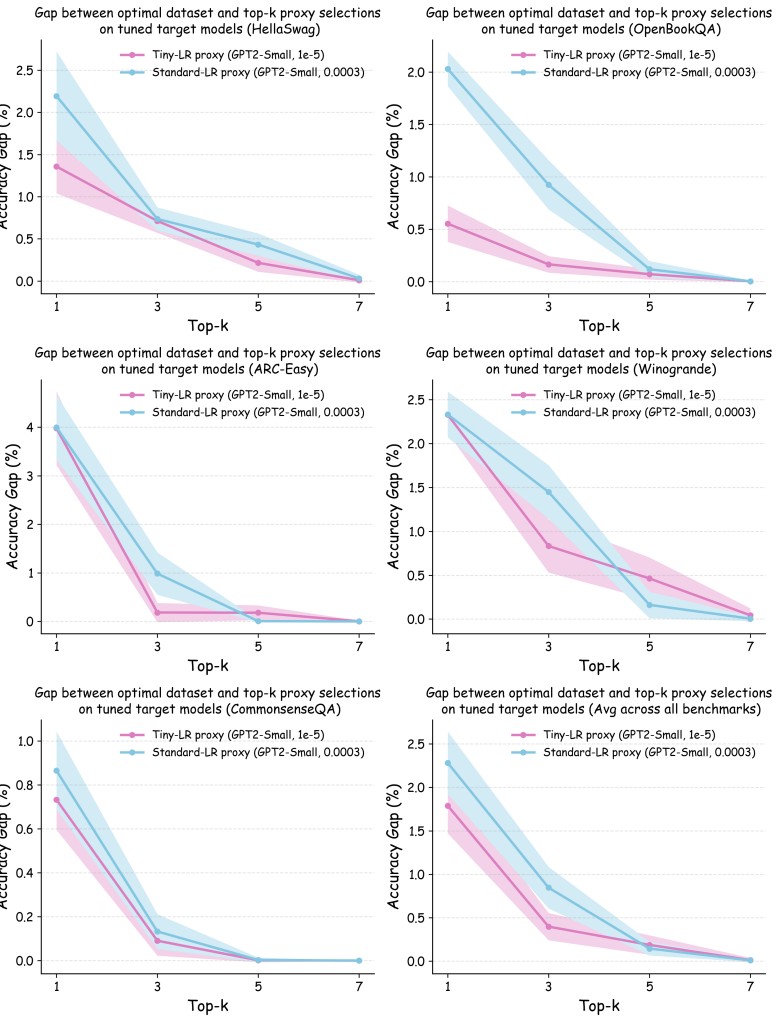

Figure 17: Top-$k$ decision regret between the optimal dataset and the best dataset among top-$k$ proxy (GPT2-Small) selections on hyperparameter-optimized target models (Pythia-1B), measured using downstream benchmarks.

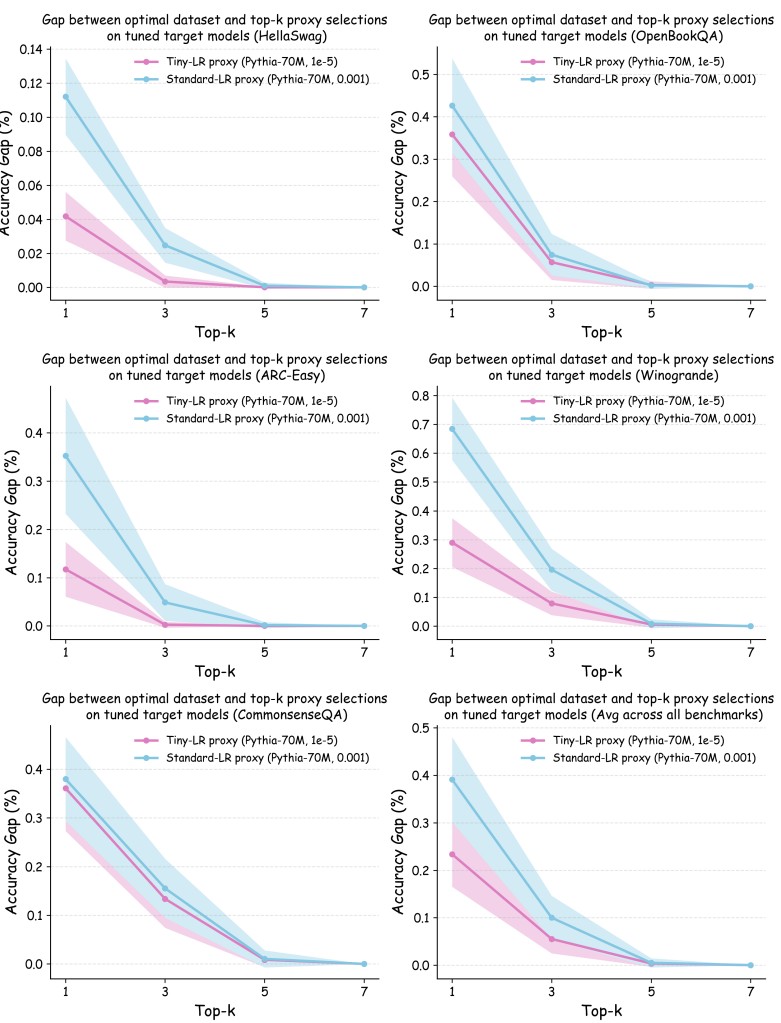

Figure 18: Top-$k$ decision regret between the optimal dataset and the best dataset among top-$k$ proxy (Pythia-70M) selections on hyperparameter-optimized target models (Pythia-1B), measured using downstream benchmarks.

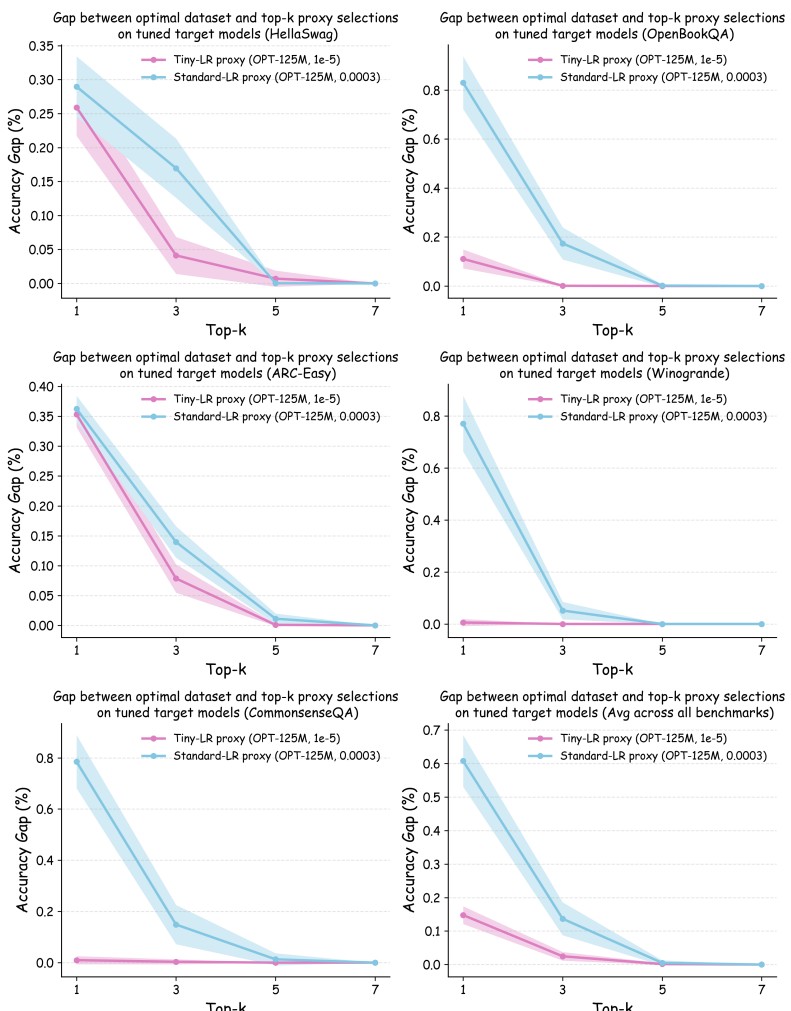

Figure 19: Top-$k$ decision regret between the optimal dataset and the best dataset among top-$k$ proxy (GPT2-Small) selections on hyperparameter-optimized target models (Pythia-1B), measured using downstream benchmarks.

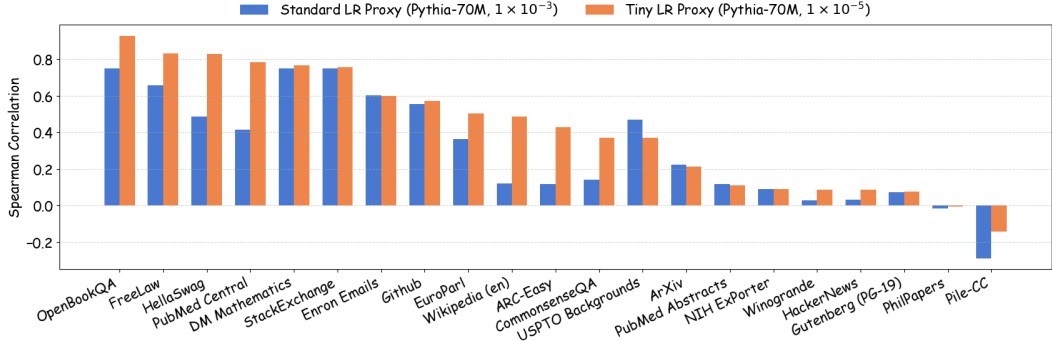

Figure 20: Average rank correlation between proxy (Pythia-70M) and target (Pythia-1B) for the loss computed over a variety of validation domains (from Pile) and downstream benchmarks.

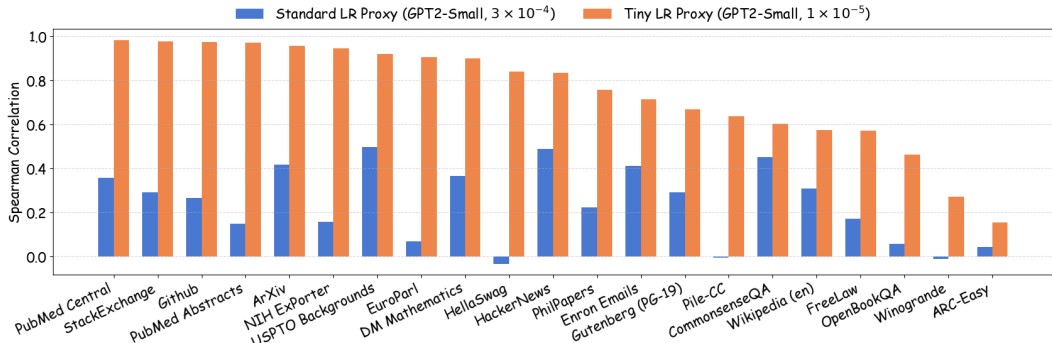

Figure 21: Average rank correlation between proxy (GPT2-Small) and target (Pythia-1B) for the loss computed over a variety of validation domains (from Pile) and downstream benchmarks.

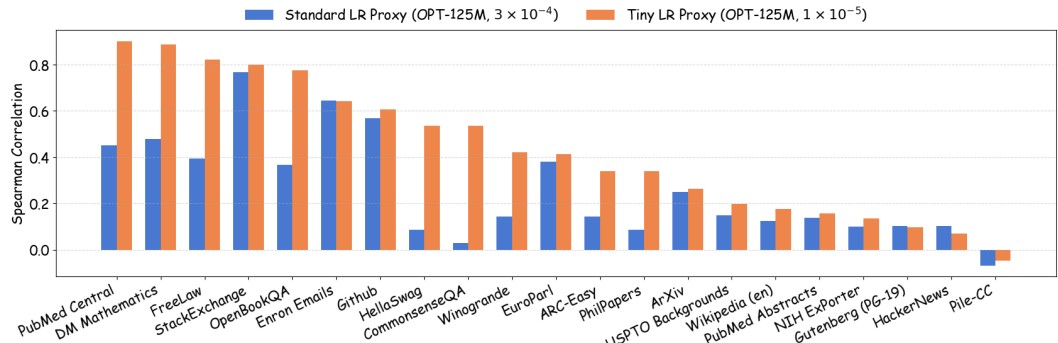

Figure 22: Average rank correlation between proxy (OPT-125M) and target (Pythia-1B) for the loss computed over a variety of validation domains (from Pile) and downstream benchmarks.

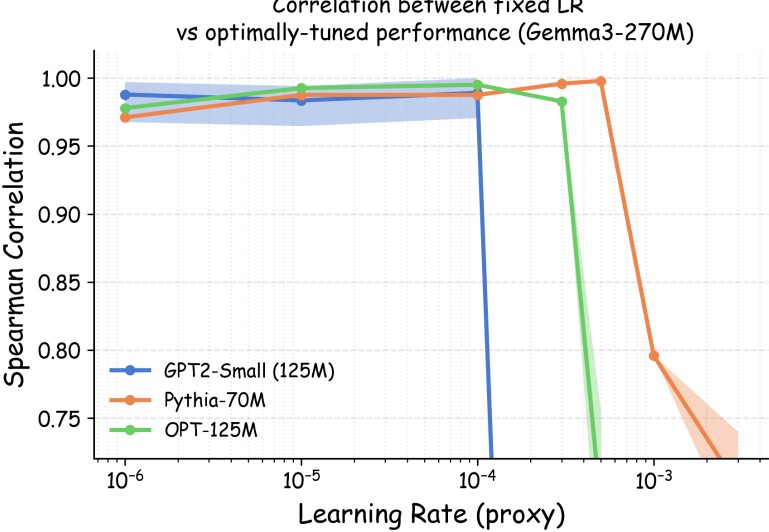

Figure 23: Spearman rank correlation between dataset rankings from proxy models (GPT2-Small, Pythia-70M, OPT-125M) and larger target model (Gemma3-270M) as a function of proxy model learning rate, evaluated on aggregated Pile validation loss. Shaded areas show 95% bootstrap CIs computed by resampling seeds (3 runs per data recipe).

### D.2.2 ABLATION STUDIES ON OTHER HYPERPARAMETERS

Figure 24, 25, and 26 conduct ablation studies to examine the transferability of small proxy models trained with varying batch sizes, weight decay coefficients, and token-per-parameter ratios (TPP), again with respect to the optimally-tuned large target models. All experiments maintain our single-epoch training protocol with no sample repetition.

**Batch size and weight decay.** As illustrated in Figure 24 and 25, we systematically evaluated batch sizes spanning $\{32, 64, 128, 256\}$ and weight decay coefficients $\{0.001, 0.01, 0.1, 1.0\}$ (covering the typical range used in LLM pretraining). The results demonstrate that dataset rankings maintain Spearman correlations above 0.90 across all configurations when small learning rates are used. This stands in sharp contrast to the dramatic ranking reversals observed with learning rate variations (correlation drops below 0.75 at standard learning rates). These findings justify our focus on learning rate as the primary factor affecting proxy model transferability.

**Token-per-parameter ratio (TPP).** Due to computational constraints, the TPP ablation study employed Pythia-410M as the target model and was limited to 14 data recipes, as each recipe required dataset-specific hyperparameter optimization. Figure 26 examines TPP ratios ranging from 20 (Chinchilla optimal) to 160 ($8\times$ Chinchilla optimal). Across this range, dataset rankings maintain high correlation with the target model's optimal rankings (Spearman $\rho > 0.85$), demonstrating robustness to overtraining. We attribute this stability to the one-epoch convention of LLM pretraining, which eliminates sample repetition and thus avoids the overfitting dynamics that could otherwise confound dataset comparisons.

Our comprehensive ablation studies across batch size, weight decay, and TPP ratios demonstrate that dataset rankings remain stable across these dimensions within practical training regimes, provided small learning rates are used. This validates learning rate as the critical hyperparameter for proxy model transferability and confirms that our proposed tiny learning rate approach provides robust dataset selection across the diverse training configurations encountered in practice.

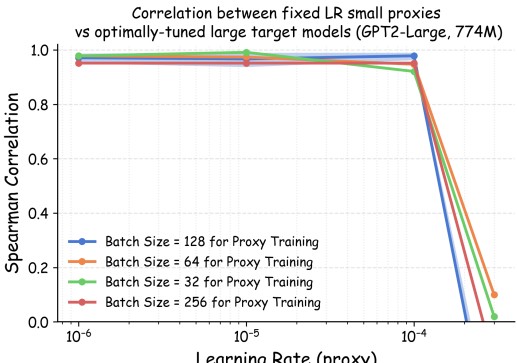

Figure 24: Spearman rank correlation between dataset rankings from proxy models (GPT2-Small) trained with different batch sizes and a larger target model (GPT2-Large) as a function of proxy model learning rate, evaluated on aggregated Pile validation loss.

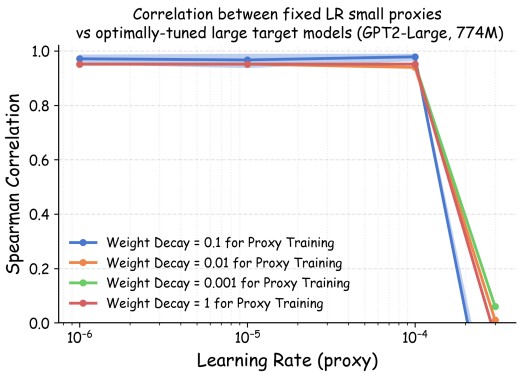

Figure 25: Spearman rank correlation between dataset rankings from proxy models (GPT2-Small) trained with different weight decay values and larger target model (GPT2-Large) as a function of proxy model learning rate, evaluated on aggregated Pile validation loss.

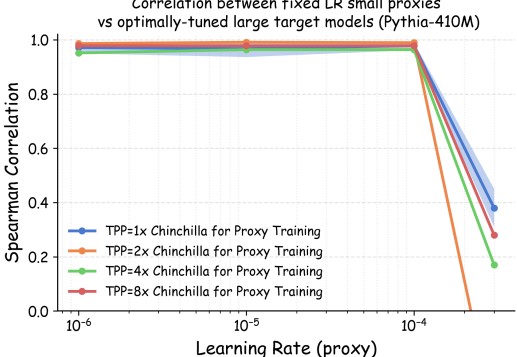

Figure 26: Spearman rank correlation between dataset rankings from proxy models (GPT2-Small) trained with different token-per-parameter ratios (TPP) and larger target model (Pythia-410M) as a function of proxy model learning rate, evaluated on aggregated Pile validation loss.

### D.2.3 Visualization of Learning Rate vs Model Performance

To further illustrate how different data recipes exhibit varying optimal learning rates, we provide detailed learning rate vs validation loss curves for the six data recipes in "scoring-based data filter" category. As shown in Figure 27, the data recipes exhibit non-trivial gaps in optimally tuned performance ($> 0.2$ in validation loss between best and worst). Furthermore, their optimal learning rates are not fixed. We can see that the data recipes with higher head_middle percentages require higher learning rates. In particular, when the learning rate is $3 \times 10^{-3}$, the 70:30 mixture appears best, but its optimally tuned performance is $> 0.05$ worse than that of the 50:50 mixture. In contrast, smaller learning rates ($< 1 \times 10^{-3}$) correctly rank these data recipes. Note that the 50:50 mixture (including substantial "tail" data) achieves optimal performance, illustrating that quality signals like perplexity are imperfect proxies (Wikipedia represents only a small portion of the Pile dataset). This underscores the critical role of proper ablation experiment protocols.

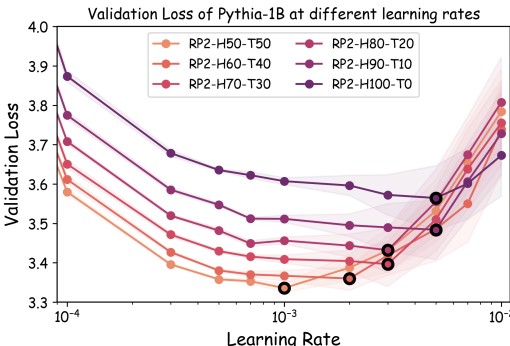

Figure 27: Validation loss curves as a function of learning rate for six RedPajama-V2 data recipes mixing head_middle and tail partitions. Each curve represents a different mixing ratio, with RP2-H50-T50 denoting 50% head_middle and 50% tail, up to RP2-H100-T0 (100% head_middle). Black circles mark the optimal learning rate for each recipe. All models are Pythia-1B trained on Pile.

### D.2.4 EMPIRICAL VERIFICATION OF GRADIENT ALIGNMENT HYPOTHESIS

In this section, we empirically validate the mechanistic hypothesis proposed in Section 5 that tiny learning rates succeed by isolating the stable first-order gradient-alignment signal in the loss reduction.

We directly measure the gradient alignment term. Specifically, we define the accumulated first-order gradient alignment score $g_{\text{align}} = \sum_{t=0}^{T} \nabla \ell_{\text{val}}(\theta_t) \cdot \nabla \ell(\theta_t; \mathcal{B}_t)$ where $\mathcal{B}_t$ is the training batch at step $t$, and $\ell_{\text{val}}$ is the loss on the validation set. This is the first-order approximation of the reduction in validation loss for vanilla SGD. In the experiment, we compute this score over the first $T = 2000$ steps for both proxy (GPT2-Small) and target (GPT2-Large) models across the 23 data recipes.

**Cross-scale stability of first-order alignment.** Figure 28 (a) shows the correlation of raw gradient alignment scores between proxy and target models. We observe near-perfect correlation ($\rho > 0.98$) across small learning rates, supporting our claim that first-order interactions are structurally stable across model scales. At standard learning rates, correlation decreases but remains reasonably high ($\rho > 0.8$). This decrease is expected because model parameters diverge more substantially when trained with larger learning rates, and since gradients depend on parameter values, this divergence naturally reduces cross-scale correlation.

**Gradient alignment vs actual performance ranking:** Figure 28 (b) shows the relationship between gradient alignment and actual performance rankings. The orange line shows the Spearman correlation between the proxy model's accumulated gradient alignment and the validation loss of the optimally-tuned large target model. Crucially, this correlation remains high ($\rho approx 0.88$) and stable even at higher learning rates, whereas the correlation based on the proxy's actual training loss (blue line) drops significantly below 0.4. We note that while highly predictive, this metric theoretically represents the update direction for vanilla SGD; since our actual training employs AdamW, which adapts updates based on moment estimation, the raw gradient alignment serves as a robust but slightly approximate proxy for the realized loss reduction.

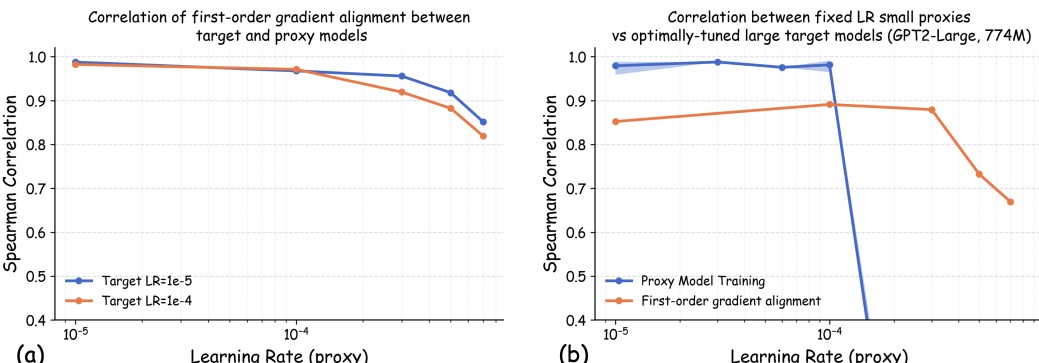

Figure 28: **Empirical verification of the gradient alignment hypothesis. (a)** Spearman correlation of accumulated gradient alignment scores between proxy (GPT2-Small) and target (GPT2-Large) models across varying learning rates. High correlation at small learning rates confirms that first-order gradient dynamics remain structurally stable across model scales. **(b)** Comparison of ranking transferability to optimally-tuned large target models: proxy model's actual validation loss (blue) versus accumulated first-order gradient alignment (orange). The gradient alignment signal maintains a strong correlation even at learning rates where actual loss rankings collapse, supporting our hypothesis that first-order terms provide a stable foundation for cross-scale dataset comparison.

### D.2.5 EFFECT OF TRAINING DURATION ON TRANSFERABILITY

We investigate whether early stopping can reliably determine dataset rankings when combined with tiny learning rates, motivated by the common practice of training for only a short period during data-ablation experiments. We measure the Spearman rank correlation between model performance at intermediate checkpoints and the optimally-tuned target model performance at the final step, using Pythia-70M with the same training configurations as our main experiments (11k total training steps) in Section 6.

**The results strongly support using early stopping when the learning rate is small.** Figure 29(a) shows that when proxy models are trained with tiny learning rates (e.g., $10^{-5}$ or $10^{-6}$), high rank correlation is maintained even with substantially fewer training steps. For instance, at 3k steps (approximately 27% of full training), tiny learning rate proxies achieve rank correlations above 0.95. This suggests that under the tiny learning rate regime, dataset rankings emerge early and remain stable, potentially enabling significant computational savings. In contrast, with standard learning rates, rank correlation generally improves as training progresses toward convergence.

**The transferability of early checkpoints depends on the learning rate schedule.** We also observe that transferability at 1k steps (mid-warmup) is notably worse than at later checkpoints across all learning rates. We hypothesize this is because the warmup phase involves rapidly changing learning rates and potentially unstable optimization dynamics. To investigate this, we conducted a pilot experiment reducing the warmup period from 2,000 to 200 steps, shown in Figure 29(b). With shorter warmup (i.e., peak learning rate reached earlier), the 1k-step checkpoint achieves comparable transferability to later checkpoints. While preliminary, this suggests that the transferability of early checkpoints depends on the learning rate schedule. We leave a systematic investigation of the interplay between early stopping and scheduler design to future work.

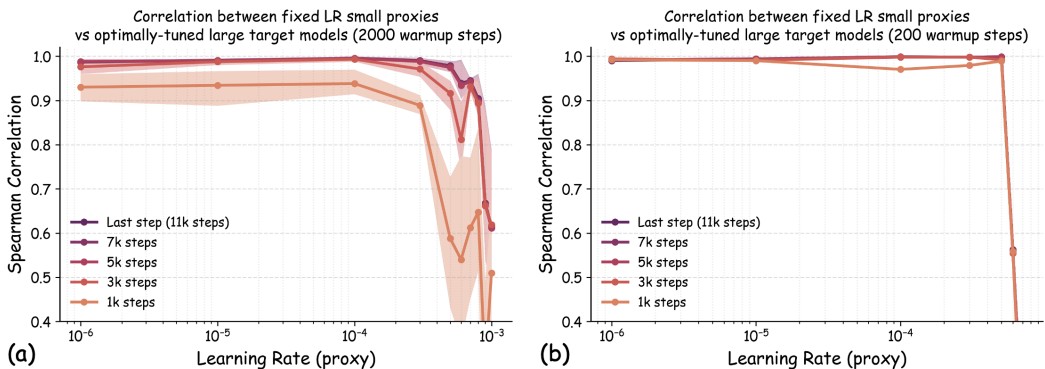

Figure 29: Effect of training duration on proxy-to-target transferability. We measure Spearman rank correlation between dataset rankings from proxy models (Pythia-70M) at intermediate checkpoints and optimally-tuned target models (GPT2-Large) at the final step. **(a)** With 2,000 warmup steps: tiny learning rates ($\leq 10^{-5}$) achieve high rank correlation early in training, while standard learning rates require more training steps for reliable transferability. **(b)** With 200 warmup steps: reducing the warmup period improves the transferability of early checkpoints.

