# OpenReview forum: "Can Small Training Runs Reliably Guide Data Curation? Rethinking Proxy-Model Practice"
_ICLR.cc/2026/Conference — ICLR 2026 Poster_

### Official Review · Reviewer_Vv1t · 2025-10-31

**Soundness:** 4
**Presentation:** 4
**Contribution:** 4
**Rating:** 8
**Confidence:** 3

**Summary:**

In the context of dataset curation, this work examines the situation where small proxy models are used to evaluate different datasets, in order to train a larger target model downstream for the best possible performance. The work finds that proxy models that are not hyperparameter-optimized are poor predictors of downstream performance. The work proposes a simple solution - train with smaller learning rates of 1-2 orders of magnitude less than standard LRs, as this makes the dataset rankings of the proxy models consistent with changes to the target model's performance given each dataset. The intuition and theory behind this solution is that small learning rates cause training to be dominated by the first order component of the gradient which optimizes for alignment of the model between the train and validation data. Using these small learning rates, experiments show that the dataset rankings given by small GPT proxy models are consistent with a) rankings of the proxy models with tuned hyperparameters, and b) rankings of a larger target LLM.

**Strengths:**

The problem of dataset curation is highly relevant, and the main result (that dataset ranking is sensitive to hyperparameters) is quite intriguing. The proposed method is simple but backed up by a theoretical arguments and experimental evidence. The paper is clearly presented and the intuition is made easy to follow.

**Weaknesses:**

The main question for me is whether the results are specific to large language transformers, or more troubling, only the GPT and Pythia models in the experiments. Given that the work already covers a lot of ground, I think instead of trying to generalize to other domains (vision especially) it would be best if the authors qualify their language and restrict their findings to the language domain specifically. As for generalizing from the specific models considered, although training is costly, it might be more convincing to have another target model that is somewhat different than the ones considered in the experiments. A weaker but cheaper experiment would be to show correlation between smaller-scale models of different types.

There is another question more for relevance: how large are the performance differences between different datasets? If they are quite small then the significance of the findings is not only diminished, but the findings themselves may again not generalize to real-world situations, e.g. if evaluating datasets with much larger differences in quality. Here I suggest a potential ablation - the authors could add some artificially bad datasets (e.g. by purposefully duplicating data) to demonstrate that their findings continue to hold outside of the narrow performance range of the datasets in this work.

Minor suggestion: figure 5 and 6 could be swapped as it seems more natural that way.

**Questions:**

How many replicates are in each of the experiments? A concern I have is whether some of the results (e.g. figure 3) are due to random chance from single runs. For some experiments this is clearly not happening, e.g. when computing correlation over multiple models or data recipes, but nevertheless it would be helpful for the authors to clarify when this may be an issue.

---

> ### Author Response · Authors · 2025-11-21
>
> We thank the reviewer for the very positive feedback and assessment!
>
> **Q1 [Scope & generalizability to other modalities]** *“Given that the work already covers a lot of ground, I think instead of trying to generalize to other domains (vision especially) ...”*
>
> **A** We sincerely thank the reviewer for this constructive and insightful suggestion! We have carefully reviewed the manuscript, qualified the language throughout the paper, and conducted pilot studies in the vision domain.
>
> **Why focus on LLM pretraining?** Proxy-model-based data recipe selection is a standard practice in frontier LLM pretraining, as evidenced by recent technical reports (e.g., Llama 3 [1], OLMo 2 [2]). We focus on this domain, where we believe our experiments will have an immediate and practical impact on both academia and industry.
>
> **Modality-agnostic theoretical analysis:** Our theoretical framework (Section 5.2) establishes guarantees for random feature models that depend only on how learning rates interact with gradient alignment and curvature terms in SGD (Theorem 2). Importantly, these theoretical results make no assumptions about the specific nature of the data modality (text, vision, etc).
>
> **Additional experiments on the vision domain.** While this work focuses on LLM pretraining, evaluating whether tiny learning rate proxy training generalizes across modalities is crucial for establishing the broader applicability of our framework. We have conducted additional pilot experiments on the vision domain, extending our approach to computer vision tasks using Vision Transformers (ViTs) on ImageNet. Specifically, we evaluated transferability between ViT-Tiny (5.7M) as the proxy model and ViT-Small (22M) as the target model. We constructed 7 distinct data recipes with varying categorical weights:
> - balanced (original ImageNet distribution)
> - animal_heavy (1.5× animals, 0.75× objects/nature), animal_light (1.2× animals, 0.9× objects/nature)
> - object_heavy (1.5× objects, 0.75× animals/nature), object_light (1.2× objects, 0.9× animals/nature)
> - nature_heavy (1.5× nature, 0.75× animals/objects), and nature_light (1.2× nature, 0.9× animals/objects)
>
> Following the same experiment protocols, we compared proxy model dataset rankings at different learning rates against target model rankings with dataset-specific hyperparameter tuning. The table below demonstrates that our core findings transfer remarkably well to the vision domain: Tiny learning rates (1e-6, 1e-5) achieve substantially higher cross-scale transferability compared to standard learning rates (3e-4 to 1e-3), despite much lower absolute validation accuracy during proxy training (as expected). These additional results on the vision domain demonstrate the **modality-agnostic nature** of our theoretical framework.
>
> **Table: ViT-Tiny => ViT-Small**
> | Learning Rate | 1e-06 | 1e-05 | 1e-04 | 3e-04 | 5e-04 | 7e-04 | 1e-03 |
> |---|---|---|---|---|---|---|---|
> | Spearman $\rho$ | 0.929 | 0.893 | 0.857 | 0.786 | 0.714 | 0.679 | 0.714 |
> | Avg Val Acc | 2.98% | 16.24% | 44.31% | 48.68% | 48.44% | 47.99% | 47.51% |
>
> We thank you again for the interesting question!
>
> [1] "The llama 3 herd of models.", 2024.
>
> [2] "2 OLMo 2 Furious" COLM 2025.
>
> **Q2 [Additional target model architecture]** *“... more convincing to have another target model ...”*
>
> **A:** We thank the reviewer for the constructive suggestion. We have conducted additional experiments during the rebuttal period using **Gemma 3 (270M)** released in March 2025 as a new target model architecture. Gemma 3 introduces several advanced architectural features absent from the proxy model architectures (GPT2, OPT, Pythia) considered in our experiments, including hybrid attention mechanism, group-query attention (GQA), QK-normalization, and the distinctive dual-placement RMSNorm design. These differences make it a strong test case for architectural generalization.
>
> The new results fully support and generalize our original conclusion. As shown in the [**figure**](https://imgur.com/a/Z1owLMS), for this relatively new transformer architecture, the tiny learning rate regime continues to yield significantly higher Spearman rank correlation (ρ > 0.9) than the standard learning rate (ρ < 0.7) across all three proxy models we considered. This demonstrates that our proposed tiny-learning-rate approach maintains its effectiveness even when transferred to more advanced transformer architectures with several different design choices. We note that our theoretical analysis in Section 5.2 is *architecture-agnostic*, which supports this empirical finding.
>
> We have included these results and the complete analysis in Appendix D.2.1.

---

> ### Author Response · Authors · 2025-11-21
>
> **Q3 [Evaluation on data recipes with artificial quality differences]** *“how large are the performance differences between different datasets ... could add some artificially bad datasets ...”*
>
> **A:** We thank the reviewer for this insightful question!
>
> **The performance differences in our experiments are substantial and practically significant.** Figure 6(b) shows that selecting data recipes using standard learning rate proxies can lead to validation loss degradation >0.25 compared to optimal choices on tuned target models—a significant difference in LLM pretraining. Figure 6(c) and additional results in Appendix D.2.1 demonstrate similar absolute performance gaps across downstream benchmarks and various target/proxy architecture configurations.
>
> **Additional plot on lr vs loss curves.** To further address the reviewer's question, we provide additional plots showing loss vs. learning rate curves for the six data recipes in the "scoring-based data filter" category, which align with the reviewer's suggestion of data recipes with "artificial quality differences" (see the side note below). As shown in the [**figure**](https://imgur.com/a/j5e0wd6), the data recipes exhibit non-trivial gaps in optimally tuned performance ($>0.2$ in validation loss between best and worst). Furthermore, their optimal learning rates are not fixed. We can see that the data recipes with higher head\_middle percentages require higher learning rates. In particular, when the learning rate is $3\times10^{-3}$, the 70:30 mixture appears best, but its optimally tuned performance is $>0.05$ worse than that of the 50:50 mixture. In contrast, smaller learning rates ($<1\times10^{-3}$) correctly rank these data recipes. Note that the 50:50 mixture (including substantial "tail" data) achieves optimal performance, illustrating that quality signals like perplexity are imperfect proxies (Wikipedia represents only a small portion of the Pile dataset). This underscores the critical role of proper ablation experiment protocols.
>
> We thank the reviewer again for the valuable question. We have incorporated the new plot and discussion into Appendix D.2.3 in the updated manuscript.
>
> **Side note: *data recipes with systematic quality variations in our experiment.*** The “scoring-based data filter” category includes six RedPajama-V2 recipes mixing "head_middle" (low perplexity) and "tail" (high perplexity) partitions in ratios from 50:50 to 100:0. These partitions are defined by the perplexity scores from 5-gram Kneser-Ney model trained on Wikipedia. Perplexity scores are a commonly used, yet controversial, signal of data quality in data filtering practices; for instance, high perplexity data could be either highly valuable or simply noisy. Hence, this represents a typical data decision scenario necessitating ablation experiments. This ambiguity makes perplexity-based filtering a representative data decision that practitioners routinely face and must evaluate through ablation experiments.
>
> **Q4 [Random seeds]** *“... whether some of the results (e.g. figure 3) are due to random chance from single runs ...”*
>
> **A:** Thanks for the catch! In the experiments shown in Figure 3, we conducted three independent training runs with different random seeds for each data recipe. The shaded regions in the figure represent the standard errors (SEs) calculated across the three replicates. We have added this detail to the caption of Figure 3.
>
> **Q5** *“Minor suggestion: figure 5 and 6 could be swapped as it seems more natural that way.”*
>
> **A:** Thanks! We have adopted this suggestion in the new manuscript.

---

> > ### Comment · Reviewer_Vv1t · 2025-11-27
> >
> > Thanks for the reply, all of my points have been addressed. I stick by my recommendation for acceptance of this paper.
> >
> > One additional question that came to mind: do the rankings stay consistent with smaller token budgets (i.e. without training to convergence)? Literature for scoring data quality will sometimes train for a short period of time as a proxy for full training - is it sufficient to do early stopping to determine the rankings of different data splits (in addition to the low learning rate)? Or from another perspective, is the speed at which the proxy model achieves some target loss (under low learning rates) correlated with the dataset rankings?

---

> > > ### Author Response · Authors · 2025-12-02
> > >
> > > We sincerely thank the reviewer again for the very positive assessment of our work!
> > >
> > > We investigated how training duration affects transferability by measuring rank correlation between model performance at intermediate checkpoints and the optimally-tuned performance at the final step. The results strongly support using early stopping when the learning rate is sufficiently small. However, we find that the optimal early-stopping point interacts with the learning rate schedule, suggesting a promising direction for future investigation.
> > >
> > > **Early checkpoints achieve strong transferability with sufficiently small learning rates.** We use Pythia-70M with the same training configurations as our main experiments (11k total training steps). As shown in [Figure](https://imgur.com/a/79p3Ts7) (a), with tiny learning rates (e.g., $10^{-5}$ or $10^{-6}$), high rank correlation (>0.95) is achieved with only 3k training steps (around 30% of full training). This suggests that reliable dataset rankings emerge early under the tiny learning rate regime, potentially leading to further compute savings in data ablation experiments. In contrast, with standard learning rates, rank correlation is significantly lower at early checkpoints and fluctuates as training progresses, suggesting that standard proxies require longer training to achieve even their limited baseline reliability.
> > >
> > > **The transferability of early checkpoints depends on the learning rate schedule.** We observe that transferability at 1k steps (mid-warmup) is notably worse than at later checkpoints. This is likely because the warmup phase involves rapidly changing learning rates and potentially unstable optimization dynamics. To test this, we conducted a pilot experiment reducing warmup from 2,000 to 200 steps, as shown in [Figure](https://imgur.com/a/79p3Ts7) (b). With a shorter warmup, the 1k-step checkpoint achieves transferability comparable to that of later checkpoints. While preliminary, this suggests that the warmup phase may confound transferability measurements. We leave a systematic investigation of the interplay between early stopping and learning rate scheduler design for future work, noting its potential to reduce ablation costs further.
> > >
> > > We have incorporated this additional result into Appendix D.2.6. Overall, the results of the pilot experiment open promising directions for future work on the interplay between the learning rate schedules and early stopping in data recipe ablation experiments. We thank the reviewer for this excellent question!

---

### Official Review · Reviewer_MjAE · 2025-10-31

**Soundness:** 2
**Presentation:** 3
**Contribution:** 2
**Rating:** 6
**Confidence:** 3

**Summary:**

The paper addresses how small "proxy" models can reliably select pretraining data for large-scale models. It identifies a key flaw in standard practice: proxies are trained with fixed hyperparameters, while large models are tuned for the selected data. The authors demonstrate this makes proxy-based dataset rankings highly fragile, even reversing with minor learning rate adjustments. The primary contribution is a new objective—to find the dataset with the best optimally-tuned performance—and a simple solution: train proxy models with a "tiny" learning rate. This "patch" is shown to be effective, achieving >0.95 rank correlation with tuned, large target models across 23 data recipes. The authors provide theoretical intuition based on first-order alignment and a proof for random feature models.

**Strengths:**

1. Clear Problem, Simple Solution: The paper identifies a relevant flaw in a common workflow (using fixed-HP proxies). The proposed solution ("use a tiny LR") is straightforward and easy to implement.

2. Helpful Empirical Validation: The paper provides a valuable set of experiments across 23 data recipes and 3 model families . Crucially, the authors use the correct (and expensive) ground truth: large models with per-dataset hyperparameter tuning. The results in Figure 2 and 5 clearly show the method's effectiveness .

3. Helpful Conceptual Reframing: The paper's reframing of the data selection objective (Section 3) is a useful perspective for the community. It correctly points out that we should optimize for a dataset's potential after tuning, not its performance in a fixed setting.

4. Supporting Ablation Studies: The appendices show that the tiny-LR method appears robust to other hyperparameter choices like batch size, weight decay, and token-per-parameter ratios, which helps isolate the learning rate as a uniquely sensitive factor .

**Weaknesses:**

1. Gap Between Theory and Claim: The paper calls its solution "theoretically-grounded", but the provided theory (Section 5.2) does not fully support its main cross-scale claim. The Random Feature Model analysis (Theorem 2) supports a within-scale claim (a proxy's tiny-LR loss predicts its own optimal loss) , but does not prove the paper's central thesis (that a proxy's loss predicts a target model's optimal loss). This disconnect undermines the claim that the solution is well-understood from first principles and makes it feel more like an (albeit effective) empirical heuristic.

2. Significant Scope Limitation (Single-Epoch): The paper's findings are restricted to single-epoch training. While this is a clean setting for a study, it is a major departure from many modern (especially Chinchilla-optimal) multi-epoch training regimes. The authors acknowledge this as future work , but it remains a significant open question whether the "tiny-LR" advantage holds when long-term optimization dynamics, data ordering, and sample repetition are in play. This limits the currently demonstrated applicability of the patch.

3. Lack of Mechanistic Evidence: The paper proposes "gradient alignment" as the intuition for why this works , but this hypothesis is not empirically tested. The paper would be much stronger if it provided direct measurements of the first-order (alignment) and higher-order terms across scales to show the former is stable while the latter is not. Without this, the "why" remains speculative, and the contribution is primarily an empirical observation.

**Questions:**

1. On the Theory Gap: The theory in Section 5.2 and Theorem 2 appears to only prove a within-scale result (that a tiny-LR RFM's ranking matches an infinite-width RFM's ranking). This supports Figure 4(a), but not the paper's main cross-scale claim (Figure 5a). Can you comment on this gap? Do you have any theoretical argument that actually bridges the proxy-to-target scale gap, or is the RFM analysis purely for intuition about the tiny-LR $\rightarrow$ optimal-loss link?

2. On Gradient Alignment: Your intuition relies heavily on "first-order gradient alignment" being preserved across model scales. Do you have more direct evidence for this? For instance, did you try to measure the gradient alignment term ($\nabla l_{val}(\theta)\cdot\nabla l_{i}(\theta)$) for different datasets at the 125M scale and the 1B scale (at initialization or after short training) and check if their ranking is also preserved? This seems like a more direct test of your hypothesis.

3. On Multi-Epoch Training: Your work is restricted to a single epoch. How do you speculate these results would change in a multi-epoch regime? One could imagine that over multiple epochs, the "higher-order" effects that you claim scramble rankings would have more time to accumulate, even with a tiny LR. Does the "patch" still work when models are trained for 2, 4, or 10 epochs? This seems like a critical question for real-world adoption.

---

> ### Author Response · Authors · 2025-11-21
>
> We thank the reviewer for the positive feedback and assessment!
>
> **Q1 [Theory for cross-scale transferability]**
>
> **A:** We thank the reviewer for their rigorous examination!
>
> We clarify that Theorem 2 **does** provide formal justification for cross-scale transferability, through the **infinite-width model** as a common "anchor" for both small proxy and large target models. The logical structure is **analogous to the Law of Large Numbers**, where two different estimators (proxy and target model) are consistent because they both converge to the same population parameter (the "*infinite-width optimal ordering*").
>
> Specifically, the reasoning proceeds as follows:
> - First, we establish the *infinite-width limit* as the "ground truth" reference for dataset ranking. This limit represents the **best achievable performance** for Random Feature Models on the training distribution. In our theoretical setup for Random Feature Models, we consider a class of "well-behaved" distributions (detailed in Appendix B.1, “Compatible with the Validation Distribution”), which satisfy the standard scaling property that the optimally-tuned validation loss $L_{\text{val-opt}}^{(m)}$ converges to this theoretical optimum $L_{\text{val}}(f^*_D)$ as the model width $m \to \infty$. This convergence property is a mild and well-justified condition that aligns with neural scaling laws. By considering well-behaved data distributions, optimally tuned large target models will rank candidate datasets according to this *infinite-width optimal ordering* (with high probability).
> - Second, Theorem 2 proves that a finite-width proxy model trained with a sufficiently tiny learning rate *also* ranks datasets according to this same *infinite-width optimal ordering*. Specifically, we prove that the sign of the loss difference for the proxy model matches the sign of the difference in $\mathcal{L}_{\text{val}}(f^*_D)$. By **combining these two convergent behaviors**, we establish the cross-scale bridge. Since the tiny-LR proxy ranking converges to the infinite-width ranking (Theorem 2), and the large-scale tuned target ranking also converges to the infinite-width ranking (by the definition of model scaling and convergence), the proxy and target rankings must align with each other.
>
> We have substantially revised Section 5.2 and Appendix B to improve the clarity of our theoretical contributions.
>
> **Q2 [Single-epoch training is standard practice?]**
>
> **A:** We appreciate the opportunity to clarify that **single-epoch training is the dominant standard for LLM pretraining,** rather than a departure from modern practices. To our knowledge, **no state-of-the-art LLMs nowadays use multiple epochs for pretraining**. The most recent frontier models, including Llama 3 [1], Qwen 3 [2], and OLMo 2 [3], strictly adhere to single-epoch regimes on massive datasets.
>
> **This single-epoch practice is directly guided by Chinchilla's scaling laws**, which demonstrate that model performance scales predictably with the number of *unique* training tokens. In particular, the Chinchilla-optimal token-to-parameter ratio (20:1) is defined with respect to *unique* tokens, making single-epoch pretraining the preferred implementation for its predictable cost-performance trade-offs.
>
> **Single-epoch pretraining is the most costly stage in LLM development.** While we acknowledge that some supervised fine-tuning and continual learning scenarios use multi-epoch training, the initial pretraining phase—the most computationally expensive stage and the primary use case for proxy-model-based methods—overwhelmingly uses single-epoch training. Given that this practice will likely remain standard for frontier model pretraining, our work targets the highest-stakes decision point in the pipeline.
>
> We thank the great question and have included the above discussion in the updated manuscript.
>
> [1] "The llama 3 herd of models." (2024)
>
> [2] "Qwen3 technical report." (2025)
>
> [3] "2 OLMo 2 Furious." (2024)

---

> ### Author Response · Authors · 2025-11-21
>
> **Q3 [Additional experiments on multi-epoch training]** *“On Multi-Epoch Training ...”*
>
> **A:** Thanks for the question! We conducted additional experiments and found that smaller learning rates continue to exhibit stronger transferability in multi-epoch settings.
>
> **Additional experiments.** We trained models for 4 epochs on datasets with 25% of the unique tokens, maintaining a constant total token-per-parameter ratio of 20 (identical to our single-epoch experiments). This design allows us to isolate the effect of data repetition while controlling for total compute. Due to methodological challenges in designing small-scale proxies for multi-epoch training (see **“Discussion and open challenges”** below), we evaluated the correlation between performance at fixed proxy learning rates and optimally tuned performance at the *same* model scale. As shown in this [figure](https://imgur.com/a/zkVwE8w), smaller learning rates continue to lead to stronger transferability in multi-epoch setting.
>
> **Discussion & open challenges.** As noted in **Q2**, multi-epoch training is *not* currently used in practice for LLM pretraining. As high-quality data becomes increasingly scarce, multi-epoch training is indeed likely to become more relevant. However, proxy-model-based techniques are not readily extended to multi-epoch settings. The core difficulty lies in designing principled subsampling strategies that properly account for complex repetition patterns across epochs. For instance, if the target model trains for 10 epochs, should the proxy model train on the same dataset for fewer epochs, or on a subset of the dataset with the same epochs? How can we ensure that the proxy's data-repetition patterns meaningfully predict the target model's learning dynamics under repetition? Similar technical challenges also arise in curriculum learning. Given the substantial materials in our current work, addressing these questions would exceed what can reasonably be covered in a conference paper. We therefore identify this as an important direction for future work and believe that our single-epoch findings provide a strong foundation for it.
>
> We thank the reviewer for the valuable question and have included this additional result in **Appendix D.2.4**.
>
> **Q4 [Mechanistic evidence]** *“Paper would be much stronger if it provided direct ...”*
>
> **A:** Thanks for this constructive suggestion! We conducted additional experiments to provide direct empirical evidence for the theoretical intuition in Section 5.1.
>
> **Setup.** We computed the *accumulated first-order gradient alignment score* $\sum_{t} \nabla \ell_{val}(\theta_t) \cdot \nabla \ell_{train}(\theta_t)$ between training and validation data over 2,000 warm-up steps for each data recipe and model scale. We use accumulated gradient alignment as it is the first-order approximation of the loss reduction in multi-step SGD.
>
> **Cross-scale stability of first-order alignment.** This [figure](https://imgur.com/a/a8f3C7o) (a) shows the correlation of raw gradient alignment scores between proxy and target models. We observe near-perfect correlation ($\rho > 0.98$) across small learning rates, supporting our claim that first-order interactions are structurally stable across model scales. At standard learning rates, correlation decreases but remains reasonably high ($\rho > 0.8$). This decrease is expected because model parameters diverge more substantially when trained with larger learning rates, and since gradients depend on parameter values, this divergence naturally reduces cross-scale correlation.
>
> **Gradient alignment vs actual performance ranking:** This [figure](https://imgur.com/a/a8f3C7o) (b) shows the relationship between gradient alignment and actual performance rankings. As we can see, the correlation between the proxy model's accumulated gradient alignment and the validation loss of the optimally tuned large model remains high ($\rho approx 0.88$) and stable even at higher learning rates. In contrast, the correlation based on the proxy's actual training loss (blue line) drops significantly below 0.4. We note that while highly predictive, this metric is derived for vanilla SGD; since the actual training uses AdamW, the raw gradient alignment serves as an approximation to the actual loss reduction.
>
> We would also like to clarify a subtlety in the one-step analysis in Section 4. Our argument is not that higher-order terms are necessarily unstable across scales. Rather, when the learning rate $\eta$ is moderately large, the $O(\eta^2)$ scaling makes higher-order terms significant enough to dominate the total loss change, thereby confounding the ranking that would otherwise be determined by first-order alignment. The empirical results above, where first-order alignment remains predictive even as full-loss rankings degrade at moderate $\eta$, validate this mechanism.
>
> We thank the reviewer for the valuable question and have included this additional result in **Appendix D.2.5**.

---

### Official Review · Reviewer_Uruq · 2025-11-01

**Soundness:** 2
**Presentation:** 3
**Contribution:** 2
**Rating:** 4
**Confidence:** 3

**Summary:**

This paper investigates the reliability of using small proxy models that use fix sets of hyperparameters to to guide data curation decisions for large-scale LLM training. The authors identify a critical fragility in current practices: dataset rankings from proxy models that rely on fixed set of hyperparameters to train on each datasets may not reveal correctly the utility of each dataset. Instead, the hyperparameters used to evaluate the utility of each dataset should be optimized for each dataset. The paper also identify that learning rate is the most important hyperparameter affecting the proxy models' dataset selection. They propose training proxy models with "tiny" learning rates (10⁻⁵ to 10⁻⁶) to improve cross-scale transferability, supported by theoretical analysis on random feature models and comprehensive experiments across 23 data recipes.

**Strengths:**

The paper is well-written. Addresses a real challenge faced by AI labs where data teams must make expensive curation decisions based on small-scale experiments. The demonstration that minor learning rate variations can completely flip dataset rankings (Figure 3) is compelling and well-presented. The tiny learning rate solution is simple to implement without requiring architectural changes or complex procedures.

**Weaknesses:**

Only single-epoch training (multi-epoch is increasingly important as data becomes scarce). I am not sure if in pratice, people only use single-epoch training to evaluate the utility of datasets. Also, of course, hyperparameter tuning on each dataset is the best to evaluate the utility of a dataset, however, it can be very costly to do in practice. Furthermore, smaller learning rate may make it slower to reach optimum point, which may affect the correct ranking of datasets if trained on single epoch. I feel this point is not addressed.

**Questions:**

Also, how are other factors such as dataset sizes, models type, and type of proxy model versus the actual model affect this ranking decision. I dont expect you to run too experiments but further discussion is appreciated.

---

> ### Author Response · Authors · 2025-11-21
>
> We thank the reviewer for the positive feedback!
>
> **Q1 [Single-epoch training as standard practice?]** *“Only … evaluate the utility of datasets”*
>
> **A:** We appreciate the opportunity to clarify that **single-epoch training is the dominant standard for LLM pretraining**. To our knowledge, **no state-of-the-art LLMs nowadays use multiple epochs for pretraining**. The most recent frontier models, including Llama 3 [1], Qwen 3 [2], and OLMo 2 [3], strictly adhere to single-epoch regimes on massive datasets. This practice is guided by Chinchilla's scaling laws, which show that model performance scales predictably with the number of *unique* training tokens, making single-epoch pretraining the preferred approach for its predictable cost-performance trade-offs.
>
> **Single-epoch pretraining is the most costly stage.** While we acknowledge that some supervised fine-tuning and continual learning scenarios use multi-epoch training, the initial pretraining phase—the most computationally expensive stage and the primary use case for proxy-model-based methods—overwhelmingly uses single-epoch training. Given that this practice will likely remain standard for frontier model pretraining, our work targets the highest-stakes decision point in the pipeline.
>
> **Broader applicability of our findings:** Following the reviewers' suggestions, we have also conducted pilot experiments to extend our findings to **multi-epoch pretraining** (in our response to *Q3 for Reviewer MjAE*) and **post-training** (in our response to *Q2 for Reviewer yFWw*). These results suggest our findings generalize beyond single-epoch pretraining and lay the groundwork for future investigations across other training paradigms.
>
> Thanks for the great question!
>
> [1] "The llama 3 herd of models." (2024)
>
> [2] "Qwen3 technical report." (2025)
>
> [3] "2 OLMo 2 Furious." (2024)
>
>
> **Q2 [Clarification on training duration with small learning rates]** *"smaller learning rate may make ... if trained on single epoch."*
>
> **A:** We appreciate the reviewer raising the critical questions regarding the compute cost.
>
> **Small learning rates do not require longer training time in our experiments.** In the standard LLM pretraining paradigm, models are trained for a **predetermined number of tokens**, not until convergence. In our experiments, all models are trained using a fixed token budget defined by *Chinchilla compute-optimal* token-per-parameter ratio ($\text{TPP}=20$). Whether using standard or tiny learning rates, every proxy model trains on the same number of tokens. Therefore, the compute budget is identical to the existing fixed-hyperparameter proxy training practices; the only change is the learning rate in our proposed approach.
>
> **Discussion on convergence and ranking.** The reviewer is correct that a smaller learning rate slows down convergence. However, **achieving an optimal performance score during proxy training is not the objective**. What matters is rank consistency—whether the relative ordering of datasets at the proxy scale reliably predicts the relative ordering of their optimally achievable performance at the large target scale. Figure 4(a) demonstrates this: the validation losses from tiny learning rate training (x-axis) are significantly higher than the optimally-tuned losses (y-axis), but the strong correlation ($\rho \approx 0.97$) demonstrates that relative ranking is preserved. We provide high-level intuition for this phenomenon through the lens of gradient alignment in Section 5.1 and offer a formal proof for random feature models in Section 5.2. Additional experiments further validating our theory are detailed in Appendix D.2.5.
>
> **Q3 [Hyperparameter tuning is costly?]** *"hyperparameter tuning … costly to do in practice."*
>
> **A:** We fully agree that per-dataset hyperparameter tuning is prohibitively expensive—*this is precisely our motivation*. Our key finding is that training proxy models with a tiny learning rate produces dataset rankings that strongly correlate with those from full per-dataset tuning. In other words, practitioners can identify the best-performing data recipe **without actually performing expensive tuning** on each candidate data recipe.

---

> ### Author Response · Authors · 2025-11-21
>
> **Q4 [Discussion: dataset sizes & model types]** *“how are other factors such as dataset sizes, models type, and type of proxy model versus the actual model affect this ranking decision.”*
>
> **A:** We thank the reviewer for this important question.
>
> **Dataset sizes (token-per-parameter ratios):** Our main experiments use the Chinchilla-optimal token-per-parameter (TPP) ratio of 20, as detailed in Section 6.1. To examine the robustness of our approach to varying dataset sizes, we conducted ablation studies across TPP ratios ranging from 20 to 160 (8× Chinchilla-optimal), presented in Figure 26 in Appendix D.2.2. Dataset rankings remain remarkably stable across this range when using tiny learning rates, with Spearman rank correlation between proxy models and optimally tuned target models remaining above 0.90 across all tested configurations. We attribute this stability to the one-epoch convention of LLM pretraining, which eliminates sample repetition and thus avoids the overfitting dynamics that could otherwise confound dataset comparisons.
>
> **Model architectures (proxy vs target):** We conducted comprehensive experiments across diverse model families to ensure our findings generalize beyond specific architectural choices. Our evaluation includes three proxy model architectures (Pythia-70M, GPT2-Small, OPT-125M) and two target model architectures (Pythia-1B, GPT2-Large), deliberately testing cross-family transferability where proxy and target models belong to different families. Figure 6(a) in the main paper directly addresses the cross-family transferability, demonstrating a target-centric view where GPT2-Large is the hyperparameter-tuned reference. All three proxy architectures, including the cross-family Pythia and OPT models, achieve a strong rank correlation ($\rho>0.92$) with the GPT2-Large target when trained in the tiny learning rate regime. **Experiments on additional target model architecture:** Following the suggestions by Reviewer Vv1t, during the rebuttal period, we further validated our approach on Gemma-3-270M (released March 2025), which features advanced architectural innovations absent from the proxy models considered in the original experiments (hybrid attention, group-query attention, QK-normalization, and dual-placement RMSNorm). As shown in the [figure](https://imgur.com/a/Z1owLMS), even with these substantial architectural differences, tiny learning rates maintained strong transferability ($\rho>0.9$) across all proxy models. We have incorporated this additional plot in Appendix D.2.1. See our response to *Q2 for Reviewer Vv1t* for details.
>
> We appreciate the reviewer prompting this discussion and have incorporated it into the revised manuscript.

---

### Official Review · Reviewer_yFWw · 2025-11-01

**Soundness:** 4
**Presentation:** 3
**Contribution:** 3
**Rating:** 8
**Confidence:** 4

**Summary:**

This paper explores the reliability of small-scale experiments for predicting larger-scale training outcomes, which especially focuses on the data selection problem in the proxy model approach. A key finding is that learning rates can influence dataset rankings. To improve the accuracy of proxy model-based predictions, the paper suggests the use of sufficiently small learning rates. The paper provides both theoretical understanding and experimental evidence to support their findings.

**Strengths:**

- The paper offers valuable insights into the proxy model practice. The proposed idea (using small enough learning rates) is notably simple, which is a great advantage for real-world applications.
- The paper provides rigorous experimental results and detailed findings, which contribute to the reliability of the study.
- The paper includes various theoretical analyses that deepens the understanding of the underlying reasons of findings.

**Weaknesses:**

- While the Appendix includes some ablation studies on other hyperparameters, more explicit and detailed explanations regarding the impacts of these hyperparameters would greatly enhance the clarity and utility of the paper. It would allow readers to better understand the sensitivity and robustness of the model to different configurations.
- Further discussion on how the observed findings might change when applied to different training strategies, such as reinforcement learning vs supervised fine-tuning, could significantly broaden the paper’s scope and strengthen its overall conclusions.

**Questions:**

The major questions and suggestions are in the weaknesses section.

---

> ### Author Response · Authors · 2025-11-21
>
> We sincerely thank the reviewer for the very positive feedback and assessment!
>
> **Q1 [Discussion on the impact of other hyperparameters]** *“While the Appendix includes some ablation studies ... would greatly enhance the clarity and utility of the paper.”*
>
> **A:** We thank the reviewer for this valuable suggestion. We have revised the main paper and substantially expanded Appendix D.2.2 to provide a detailed analysis of how batch size, weight decay, and token-per-parameter (TPP) ratios affect dataset ranking stability. Overall, our ablation studies reveal that while dataset rankings can completely reverse as learning rate changes, they remain remarkably stable across practical ranges of batch size, weight decay, and token-per-parameter ratios.
>
> **Batch size and weight decay:** We systematically evaluated batch sizes spanning {32, 64, 128, 256} and weight decay coefficients {0.001, 0.01, 0.1, 1.0}, covering the ranges typically used in LLM pretraining. Our results (Appendix D.2.2, [Figures 24-25](https://imgur.com/a/1wYDIeM) demonstrate that dataset rankings maintain Spearman correlations above 0.90 across all tested configurations when small learning rates are used. This stability stands in sharp contrast to the dramatic ranking reversals observed with variations in learning rate.
>
> **Token-per-parameter ratio (TPP).** We examined TPP ratios from 20 (1× Chinchilla optimal) to 160 (8× Chinchilla optimal). The results (Appendix D.2.2, [Figures 26](https://imgur.com/a/CqIrkkO)) show that across this range, dataset rankings maintain high correlation with the target model's optimal rankings (Spearman $\rho > 0.9$), demonstrating robustness to overtraining. We attribute this stability to the one-epoch convention of LLM pretraining, which eliminates sample repetition and thus avoids the overfitting dynamics that could otherwise confound dataset comparisons.
>
> We remark that our experiments focus on hyperparameter ranges commonly used in LLM pretraining. While extreme configurations far outside these ranges (e.g., batch size of 1) will affect transferability, such settings are never used in practice. Therefore, in this work, we focus on the learning rate as the primary factor for improving proxy model transferability.

---

> ### Author Response · Authors · 2025-11-21
>
> **Q2 [Discussion on post-training paradigm]** *“Further discussion on ... strengthen its overall conclusions.”*
>
> **A:** We sincerely thank the reviewer for this constructive and insightful suggestion!
>
> **The focus on LLM pretraining.** We primarily focus on LLM pretraining as it is **the most costly stage** in model development. Also, small-scale proxy experiments are the standard approach for guiding pretraining data decisions, as documented in recent industry reports (e.g., Llama 3 [1], OLMo 2 [2]). We have carefully revised the manuscript and qualified the scope to LLM pretraining, where data decisions have the greatest impact.
>
> **Additional pilot experiments on post-training.** We conducted a new pilot experiment during the rebuttal period to evaluate the tiny learning rate method for Supervised Fine-Tuning (SFT).
> - **Setup:** We adopted a setup similar to existing literature [1], performing SFT on the Llama 3.1 8B base model using LoRA (rank 32). We constructed 10 distinct data recipes based on the Tulu 3 dataset [3] by systematically ablating (removing) one of the top 10 domain sources. Each recipe is used to fine-tune with LoRA for 500 steps with batch size 128. To isolate the effect of the learning rate on transferability, we use identical model scales for both the proxy and the target (LoRA-finetuned Llama 3.1 8B). That is, we assess the correlation between the ranking of data recipes when using a fixed, tiny learning rate and the ranking obtained when the model is optimally tuned for each recipe.
> - **Results:** We sweep learning rates from $8 \times 10^{-6}$ to $3 \times 10^{-4}$ and compute the Spearman rank correlation between rankings at each learning rate versus the optimal-HP ranking. We evaluated these recipes across three diverse benchmarks: TruthfulQA, GSM8K, and ARC-Easy. As shown in the [figure](https://imgur.com/a/KSLVedQ), dataset rankings are highly sensitive to learning rate choice in the standard regime ($1 \times 10^{-4}$ to $3 \times 10^{-4}$). However, rankings stabilize and achieve strong correlation with optimal-HP rankings when using smaller learning rates. This preliminary evidence suggests that the principle of using tiny learning rates to improve proxy-to-target transferability extends to SFT scenarios.
>
> **Discussion and open questions.** While these pilot experiments are highly encouraging, we note that a dedicated, comprehensive investigation into post-training requires addressing many open questions that exceed the scope of the current paper. For instance: **(1) Optimal proxy training design:** Unlike pretraining, post-training involves multiple proxy strategies (e.g., fewer iterations, LoRA with reduced rank, dataset subsampling). The optimal combination of these with the learning rate remains an open question. **(2) Pretraining data dependencies:** The base model's pretraining distribution may significantly influence how fine-tuning data recipes transfer across scales, particularly for domain-specific data. **(3) Multi-epoch dynamics:** Post-training typically involves multiple epochs, introducing dynamics of sample repetition and potential overfitting, which differs fundamentally from our single-epoch pretraining analysis. **(4) Curriculum effects:** Post-training often employs curriculum learning or careful data orderings, which presents a challenge for designing transferrable proxy training protocols.
>
> Given the substantial scope of these questions, we believe each merits dedicated investigation beyond what can be addressed in this paper. We have added a sentence to the Conclusion section to explicitly include the generalization to post-training dynamics as a key avenue for future research.
>
> [1] Dubey, Abhimanyu, et al. "The llama 3 herd of models." (2024)
>
> [2] OLMo, Team, et al. "2 OLMo 2 Furious." (2024).
>
> [3] Lambert, Nathan, et al. "Tulu 3: Pushing frontiers in open language model post-training." (2024).
>
> [4] Li, Yuan, Zhengzhong Liu, and Eric Xing. "Data mixing optimization for supervised fine-tuning of large language models." ICML 2025

---

### Author Response · Authors · 2025-12-03
**Summary of key updates before Nov 28**

Dear Area Chair,

To facilitate your assessment, we provide a concise summary of our paper revisions and key updates from the discussion period (through Nov 27).

We thank all reviewers for their constructive feedback. We are pleased that our work received very positive evaluations. We have carefully addressed all reviews and made substantial modifications to strengthen the paper. All changes are highlighted in blue.

**Additional experiments and major revisions:**

- **Post-training experiments (Reviewer yFWw):** Added pilot experiments extending our proxy method to SFT settings.
- **Multi-epoch pretraining (Reviewers Uruq and MjAE):** Extended our proxy method to multi-epoch pretraining settings.
- **Empirical support for theoretical explanation (Reviewer MjAE):** Provided direct empirical evidence supporting the theoretical analysis in Section 5.1.
- **Theory interpretation (Reviewer MjAE):** Revised the interpretation of our theoretical results in Section 5.2 for improved clarity.
- **New target architecture (Reviewer Vv1t):** Added experiments using the recently released Gemma 3 (270M) as a target model.
- **Vision modality generalization (Reviewer Vv1t):** Extended experiments to the vision domain using Vision Transformers (ViTs) on ImageNet.
- **Loss vs. learning rate analysis (Reviewer Vv1t):** Added detailed loss vs. learning rate curves for "scoring-based data filter" recipes.
- **Training duration effects (Reviewer Vv1t):** Extended experiments to measure the correlation between intermediate checkpoint performance and final optimally-tuned performance.

**Key updates during discussion period:**

- **Nov 25:** Reviewer Uruq raised their overall rating from 4 to 6, with improved scores for both "Soundness" and "Contributions" (each increased to 3).
- **Nov 27:** Reviewer Vv1t confirmed all questions were addressed and maintained the rating at 8, and raised an additional question regarding intermediate checkpoints for ablation experiments, which we addressed with further experiments.

**Ratings on Nov 28: 8, 6, 6, 8**

We sincerely thank the reviewers and Area Chair for their time and constructive feedback throughout this process.

Best regards,
The Authors

---

### Meta-Review · Area_Chair_xDyC · 2025-12-31

**Summary:**

The authors consider the reliability problem of proxy-model-based data curation, which is practically used in large-scale language model pretraining. The key insight is that dataset rankings obtained from proxy models trained with fixed hyperparameters can be highly unstable. The authors argue that the mismatch arises since proxy evaluations are done with fixed hyperparameters while target models are evaluated for each dataset after hyperparameter tuning. The proposed approach, i.e., training proxy models with sufficiently small learning rates, is simple yet practical. The authors provide extensive empirical evidences to support the claim. The authors also derive theoretical intuition and analysis based on first-order gradient alignment and random feature models.

The Reviewers' opinions are generally positive on the submission. The Reviewers agree that the submission is practical relevance, clear empirical findings, low-cost proposed approach. However, the Reviewers also have concerns on theoretical generality and scope limitations, e.g., single-epoch pretraining. Overall, we think although there are still some remaining limitations in theoretical depth and generality, the finding results are appealing for the community, supported by strong empirical evidences.

**Reviewer Concerns:**

The Reviewers have some following concerns:

+ Reviewer yFWw: sensitivity and robustness of the model to different configurations; different training strategies.

+ Reviewer Uruq: limitation to single-epoch training; costly of hyperparameters tuning, optimization with small learning rate;

+ Reviewer MjAE: gap between theoretical finding results and the claim; limitation of theoretical finding results w.r.t. random feature models; significant limitation scope for single-epoch setting: whether the proposed learning with small learning rate is still effective for multi-epoch settings; multi-epoch regime settings?

+ Reviewer Vv1t: limitation of experimental settings: large language transformers (GPT/Pythia); magnitude of empirical performance differences across datasets;

**Reviewer Scores:**

The authors provide several additional empirical evidences in the rebuttal; clarify the single-epoch training w.r.t. Chinchilla's scaling laws. We think the rebuttal of the authors has addressed several raised concerns from the Reviewers. The Reviewers agree that the finding results are interesting for the community.

---

### Decision · Program_Chairs · 2026-01-26

Accept (Poster)